



# Heavy air pollution with the unique "non-stagnant" atmospheric boundary layer in the Yangtze River Middle Basin aggravated by regional transport of PM2.5 over China

Chao Yu[1,2], Tianliang Zhao[1,*], Yongqing Bai[3,*], Lei Zhang[1,4], Shaofei Kong[5], Xingna Yu[1], Jinhai He[1], Chunguang Cui[3], Jie Yang[1], Yinchang You[1], Guoxu Ma[1], Ming Wu[1], Jiacheng Chang[1]

Collaborative Innovation Center on Forecast and Evaluation of Meteorological Disasters, Key Laboratory for Aerosol-Cloud-Precipitation of China Meteorological Administration, Nanjing University of Information Science and Technology, Nanjing 210044, China
Southwest Electric Power Design Institute Co., Ltd of China Power Engineering Consulting Group, Chengdu, 610021, China
Institute of Heavy Rain, China Meteorological Administration, Wuhan, 430205, China
Chengdu Academy of Environmental Sciences, Chengdu, 610031, China
Department of Atmospheric Sciences, School of Environmental Studies, China University of Geosciences (Wuhan), 430074, Wuhan, China

*Correspondence*: Tianliang Zhao (tlzhao@nuist.edu.cn); Yongqing Bai (2007byq@163.com)

**Abstract:** Regional transport of air pollutants controlled by both emission sources and meteorological factors results in a complex source-receptor relationship of air pollution change. Wuhan, a metropolis in the Yangtze River Middle Basin (YRMB) of central China experienced



heavy air pollution characterized by excessive PM$_{2.5}$ concentrations reaching 471.1 μg m$^{-3}$ in
January 2016. In order to investigate the regional transport of PM$_{2.5}$ over China and the
meteorological impact on wintertime air pollution in the YRMB area, observational
meteorological and other relevant environmental data from January 2016 were analyzed. Our
analysis presented the noteworthy cases of heavy PM$_{2.5}$ pollution in the YRMB area with the
unique "non-stagnant" meteorological conditions of strong northerly winds, no temperature
inversion and additional unstable structures in the atmospheric boundary layer. This unique set of
conditions differed from the stagnant meteorological conditions characterized by near-surface
weak winds, air temperature inversion, and stable structure in the boundary layer observed in
heavy air pollution over most regions in China. The regional transport of PM$_{2.5}$ over
central-eastern China aggravated PM$_{2.5}$ levels present in the YRMB area, thus demonstrating the
source-receptor relationship between the originating air pollution regions in central-eastern China
and the receiving YRMB regions. Furthermore, a backward trajectory simulation using
FLEXPART-WRF to integrate the air pollutant emission inventory over China was used to explore
the patterns of regional transport of PM$_{2.5}$ governed by the strong northerly winds in the cold air
activity of the East Asian winter monsoon over central-eastern China, which contributes markedly
to the heavy PM$_{2.5}$ pollution in the YRMB area. It was estimated that the regional transport of
PM$_{2.5}$ of non-local air pollutant emissions could contribute more than 65% of the PM$_{2.5}$
concentrations to the heavy air pollution in the YRMB region during the study period, revealing
the importance of the regional transport of air pollutants over central-eastern China in the
formation of heavy air pollution over the YRMB region.
**Key words:** PM$_{2.5}$ pollution; Yangtze River Middle Basin; meteorological condition; regional



transport; FLEXPART-WRF

## 1. Introduction

Air pollution events with excessive ambient PM$_{2.5}$ concentrations have been observed
frequently in the central-eastern regions of China in recent years. These events result in serious
environmental problems with adverse influence on traffic, human health, climate change and other
significant aspects (Fuzzi et al., 2015; An et al., 2019; Nel, 2005). Based on the observations in
China, there is a well-established association between haze pollution and high concentrations of
PM$_{2.5}$ (particulate matter with an aerodynamic diameter less than 2.5 μm). Air pollution levels
are highly dependent on emissions of air pollutants and changes in meteorology (Tie et al., 2017;
Xu et al., 2016b; An et al., 2019; Xu et al., 2016a). The accumulation, maintenance and dissipation
of haze pollution events are generally determined by meteorological changes (Kan et al., 2012),
among which the boundary layer structures play the most important role (Wu et al., 2017).
Meteorological conditions of stagnation characterized by near-surface low winds, high humidity
and stable boundary layer could govern the periodic variations of haze pollution, which present as
typical wintertime air pollution in central-eastern China (Xu et al., 2016b; Zhang et al., 2014;
Huang et al., 2018). Four major regions exhibiting haze pollution with high PM$_{2.5}$ concentrations
and overall poor air quality are centered over North China Plain (NCP), Yangtze River Delta
(YRD) in East China, Pearl River Delta (PRD) in South China and Sichuan Basin (SCB) in
Southwest China (Cheng et al., 2008; Zhang et al., 2012; Deng et al., 2011; Wang et al., 2016; Tie
et al., 2017; Qiao et al., 2019).



The source-receptor relationship describes the impacts of emissions from an upwind source
region to pollutant concentrations or deposition at a downwind receptor location.    Regional
transport of source-receptor air pollutants is generally complicated by two types of    factors:
emission and meteorology. The emission factor includes the emission source strength, chemical
transformation and production; the meteorological factor determines the transport pathway from
the source to receptor regions, exchanges between boundary layer and free troposphere, the
removal processes occurring over the source and receptor regions as well as along the transport
pathways. Regional transport of air pollutants with the source-receptor relationship is an important
issue in our understanding of changes in air quality. Driven by atmospheric circulation, the
regional transport of $PM_{2.5}$ from source regions can deteriorate air quality in the downwind
receptor regions, leading to the regional haze pollution observed in a large area over
central-eastern China (Chang et al., 2018; Wang et al., 2014; He et al., 2017; Chen et al., 2017b;
Hu et al., 2018; Jiang et al., 2015). The Yangtze River Middle Basin (YRMB) in central China is
geographically surrounded by four major haze pollution regions in all directions with NCP to the
north, the YRD to the east, the PRD to the south and the SCB to the west (Fig.1 a). Due to this
specialized location of the YRMB as a regional air pollutant transport hub with subbasin
topography (see Fig. 1b), the regional transport of air pollutants driven by the cold air activity of
East Asian winter monsoonal winds in central-eastern China could develop a source-receptor
relationship between major haze pollution regions (NCP, YRD, etc.) in central-eastern China and
the downwind YRMB region. However, there are unresolved questions regarding the
meteorological processes involved in the regional transport of air pollutants    and the pattern of
regional transport with contribution to the air quality changes observed in the YRMB.



Wuhan, a metropolis located in the YRMB, has confronted the problems associated with urban air
pollution, especially heavy PM$_{2.5}$ pollution events that occur in the winter (Zhong et al., 2014;
Gong et al., 2015; Xu et al., 2017; Tan et al., 2015). Local emissions of air pollutants from urban
transportation, industrial exhaust and bio-combustion play an important role in YRMB urban air
pollution (Acciai et al., 2017; Zhang et al., 2015). Many observational and modeling studies on air
pollution in this urban area have been conducted (Zheng et al., 2019; Wu et al., 2018). However,
regional transport routes of PM$_{2.5}$ from central-eastern China and its contribution to air pollution
over the YRMB are still poorly understood, especially in relation to heavy air pollution episodes in
the YRMB area. This study selected the Wuhan area as a representative area within the YRMB for
investigation of the meteorological conditions of air pollution events in January 2016 and the
contribution of regional transport of PM$_{2.5}$ to heavy air pollution over the YRMB region.
**2. Observational analysis**
**2.1 Data**
Wuhan, the capital of Hubei province, is located across the Yangtze River, where its surrounding
water network attributed with a humid environment**.** (see Fig. 1b). In order to analyze the air
quality change, the hourly concentrations of air pollutants including PM$_{2.5}$ in January 2016 were
collected from sites over central-eastern China, including ten observational sites in Wuhan. These
ten sites include nine urban sites in residential and industrial zones as well as one suburban site
within the China National Air Environmental Monitoring Network. The concentrations of air
pollutants were distributed spatially in less difference over the suburban and urban sites with the
similar patterns and peaks of hourly changes during the heavy pollution events, demonstrating the



regional heavy air pollution in a large area of the YRMB region with the contribution of regional
transport from central-eastern China, while the obviously differences in air pollutant
concentrations were measured with the relative high and low $PM_{2.5}$ concentrations respectively at
urban sites and suburban site during the clean air period, reflecting the important influence of high
air pollutant emission over urban area on local air quality. The $PM_{2.5}$ concentrations averaged over
the ten observational sites were used to characterize the variations of air pollution in January 2016
over this urban area within the YRMB.

The meteorological data of surface observation and air sounding in Wuhan and other

observatories in central-eastern China were obtained from the China Meteorological Data Sharing
Network (http://data.cma.cn/). Meteorological data selected for this study included horizontal
visibility, air temperature, relative humidity, air pressure, and wind speed and direction with
temporal resolutions of 3 h for surface observation and 12 h for sounding observation in order to
analyze the variations of the meteorological conditions in the atmospheric boundary layer in
January 2016.

The ERA (ECMWF ReAnalysis) -Interim reanalysis data of meteorology from the

ECMWF          (European          Centre          for          Medium-Range          Weather          Forecasts)
(https://www.ecmwf.int/en/forecasts/datasets/reanalysis-datasets/) were applied to explore the cold
air activity of East Asian winter monsoonal winds in January 2016 and their anomalies during
heavy $PM_{2.5}$ pollution the over central-eastern China .
**2.2 Variations in $PM_{2.5}$ concentrations and meteorology in January, 2016**
Based on the National Ambient Air Quality Standards of China released by the Ministry of



Ecology and Environment of China in 2012 (http://www.mee.gov.cn/), light and heavy air
pollution levels of $PM_{2.5}$ are categorized by daily average $PM_{2.5}$ concentration exceeding 75μg m$^{-3}$
and 150μgm$^{-3}$ in ambient air, respectively. The daily variations of $PM_{2.5}$ concentrations over
January 2016 in Wuhan are illustrated in Figure 2a. The average monthly $PM_{2.5}$ concentration
reached 105.8μg m$^{-3}$. The national secondary standard was exceeded on 27 days with daily $PM_{2.5}$
concentrations exceeding 75μg m$^{-3}$ during the entire month of January 2016 in Wuhan, indicating
that this urban area in the YRMB was suffering under significant $PM_{2.5}$ pollution during this
period. As shown in Figure 2a, a 21-day prolonged air pollution event resulted from high levels of
daily $PM_{2.5}$ concentrations (>75μg m$^{-3}$) over the period of January 1 to 21. During this 21-day
period of air pollution, three notably heavy air pollution events occurred on January 4, 10-12 and
18 with excessive daily $PM_{2.5}$ concentrations (>150μg m$^{-3}$); these events are marked as P1, P2 and
P3 in Figure 2. Based on the observation in January 2016, we found the interesting phenomenon
of an apparent 7-day cycle of heavy air pollution in January 2016, reflecting an important
modulation of meteorological oscillation in the East Asian winter monsoon affecting air pollution
concentrations observed over the YRMB region (Xu et al., 2016a). A period analysis on long-term
observation data of air quality could provide more information on air pollution oscillations with
meteorological drivers.
Figure 2b presents the hourly changes of $PM_{2.5}$ concentrations for the three heavy air pollution
events P1, P2 and P3. The heavy pollution event P1 on January 4 started at 11:00 am (local time is
used for all events) and ended at 11:00 pm at same day. with an observed $PM_{2.5}$ concentration peak
of 471.1μg m$^{-3}$. The event P2 occurred from 10:00 pm on January 10 to 00:00 a.m. on January 12
with a duration of 26 h and two peaks in $PM_{2.5}$ concentrations of 231.4μg m$^{-3}$ and 210.6μg m$^{-3}$.





The event P3 was observed between 7:00 p.m. on January 17 and 2:00 pm on January 18 with an
explosive growth rate of 42.9μg m$^{-3}$ h$^{-1}$ in PM$_{2.5}$ concentrations. Those three heavy PM$_{2.5}$ pollution
episodes over the YRMB region were characterized by short durations of less than 26 h from rapid
accumulation to fast dissipation.
Using the environmental and meteorological data observed in Wuhan in January 2016, the effects
of the meteorological conditions on PM$_{2.5}$ concentrations in the YRMB region were statistically
analyzed in regards to hourly variations of surface PM$_{2.5}$ concentrations, near-surface wind speed
(WS) and direction (WD), as well as surface air temperature (T), air pressure (P) and relative
humidity (RH) (Fig. 3). Among the observed hourly changes in PM$_{2.5}$ concentrations and
meteorological elements shown in Figure 3, the obvious positive correlations to surface air
temperature and relative humidity, as well as a pronounced negative correlation to surface air
pressure and a weak positive correlation to near-surface wind speed were found with the change of
PM$_{2.5}$ concentrations in January 2016 (Table 1). The near-surface wind speed associated with East
Asian monsoons has significantly influence concentrations of air pollutants mainly by the changes
in weak advection of cold air, in conjunction with strong subsidence and stable atmospheric
stratification, can easily produce a stagnation area in the lower troposphere resulting in regional
pollutant accumulations, which are favorable for the development of CEC haze events. In addition,
in the presence of high soil moisture, strong surface evaporation results in increases in the
near-surface relative humidity, which is also conducive to hygroscopic growth of participles for
haze formation; high air temperature and strong solar radiation could enhance chemical reactions
and conversions for the formation of secondary aerosols in the atmosphere, precipitation could
alter the emissions, and depositions of air pollutants. These observations could reflect the special



influences of meteorological factors (winds, air temperature, humidity, precipitation etc) on
physical and chemical processes in the ambient atmosphere, in particular that of wind driving air
pollutant transport and affecting air quality change in the YRMB region.

When we focused on the changes leading to excessive PM$_{2.5}$ levels during these heavy air

pollution events, it is noteworthy that all three heavy pollution episodes P1, P2 and P3 were
accompanied with strong near-surface wind speeds in the northerly direction, as well as evident
turning points in prevailing conditions leading to falling surface air temperatures and increasing
surface air pressure (noted as a rectangle with red dashed lines in Fig. 3). The conditions observed
during these three heavy pollution episodes reflect the typical meteorological characteristics of
cold front activity over the East Asian monsoon region. The southward advance of a cold front
could drive the regional transport of air pollutants over central-eastern China (Kang et al., 2019).
Climatologically, a strong northerly wind, low air temperature and high air pressure are typical
features of an incursion of cold air during East Asian winter monsoon season in central-eastern
China, which could disperse air pollutants and improve air quality in the NCP region (Miao et al.,
2018;Xu et al., 2016b). Compared to the meteorological conditions for stagnation with weak
winds observed for heavy air pollution events in the major air pollution regions of central-eastern
China (Huang et al., 2018;Ding et al., 2017), meteorological conditions with strong near-surface
wind were anomalously accompanied with the intensification of PM$_{2.5}$ during heavy air pollution
periods over the study area in the YRMB in January 2016 (Fig. 3). This could imply the
importance of regional air pollutant transport in worsening air quality over the YRMB, driven by
the strong northerly winds of the East Asian winter monsoon over China.



## 2.3 A unique "non-stagnation" meteorological condition for heavy PM$_{2.5}$ pollution

To further investigate the connection of meteorological elements in the near-surface layer with changes in air quality affected by PM$_{2.5}$ concentrations in the YRMB region, we carried out a more detailed correlation analysis of PM$_{2.5}$ concentrations in Wuhan with near-surface wind speed and air temperature and three different levels of PM$_{2.5}$ concentrations: clean air environment (PM$_{2.5}$<75μg m$^{-3}$), light air pollution (75μg m$^{-3}$ ≤ PM$_{2.5}$ <150μg m$^{-3}$) and heavy air pollution (PM$_{2.5}$ ≥150μg m$^{-3}$) periods (Table 2). As seen in Table 2, the surface PM$_{2.5}$ concentrations were positively correlated with air temperature, as well as negatively correlated with wind speeds during the periods of clean air environment and light air pollution. It should be emphasized here that a significantly negative correlation (R=-0.19) of PM$_{2.5}$ concentrations with near-surface wind speeds for the light air pollution period could indicate that weak winds are favorable for local PM$_{2.5}$ accumulation, reflecting an important effect of local air pollutant emissions on light air pollution periods over the YRMB area. In January 2016, the overall wind speed of Wuhan was weak with a monthly mean value of 2.0m s$^{-1}$, which could prove beneficial to maintaining the high PM$_{2.5}$ levels in the prolonged air pollution event experienced during January 2016. However, a significantly positive correlation (R=0.41) existed between excessive PM$_{2.5}$ concentrations (PM$_{2.5}$ >150μg m$^{-3}$) and strong near-surface wind speeds during the heavy air pollution period, which was inconsistent with the stagnation meteorological conditions observed in the near-surface layer with weak winds associated with heavy air pollution in eastern China (Cao et al., 2012; Zhang et al., 2016). The meteorology and environment conditions in the YRMB region indicate the close association of heavy air pollution periods with the intensification of regional transport of





air pollutants driven by strong winds (Fig. 3, Table 2) reflecting a key role of regional air pollutant
transport in the development of the YRMB's heavy air pollution periods.
In order to clearly illustrate the impact of wind speed and direction on the $PM_{2.5}$ concentrations
associated with the regional transport of upwind air pollutants, Figure 4 presents the relation of
hourly changes in surface $PM_{2.5}$ concentrations (in color contours) to near-surface wind speed (in
radius of round) and direction (in angles of round) in Wuhan during January 2016. As can be seen
in Figure 4, strong northerly winds of the East Asian winter monsoon accompanied extremely high
$PM_{2.5}$ concentrations (>150μg m$^{-3}$) during heavy air pollution periods, including the northeast gale
that exceeded 5 m·s$^{-1}$ during the extreme heavy pollution period with excessive high $PM_{2.5}$
concentrations (>300μg m$^{-3}$) over the YRMB region. These results reveal a unique meteorological
condition of "non-stagnation" with strong winds during events of heavy air pollution over YRMB
area. Conversely, the observed $PM_{2.5}$ concentrations ranging between 75 and 150μg m$^{-3}$ for light
air pollution periods generally corresponded with low wind speed (<2m s$^{-1}$) in the YRMB region
(Fig. 4); therefore, it is the meteorological condition for stagnation characterized by weak winds
involved in the accumulation of local air pollutants that is responsible for the YRMB's light air
pollution periods. Meteorological impacts on air quality could include not only the stagnation
condition with weak winds and stable boundary layer, but also air temperature, humidity,
precipitation, atmospheric radiation etc. in close connection with atmospheric physical and
chemical processes. Therefore,    meteorological drivers of air quality change are complicated by a
series of physical and chemical processes in the atmosphere especially the formation of secondary
air pollutants in the humid air environment overlying the dense water network in the YRMB
region (see Fig. 1b), thus pointing out the need for further comprehensive study.



As shown in Figure 2a, the heavy pollution periods with the daily average $PM_{2.5}$

concentrations exceeding $150\mu gm^{-3}$ in ambient air, respectively occurred on January 4, 10-12 and

18, and the clean air periods with the daily average $PM_{2.5}$ concentrations below $75\mu gm^{-3}$ occurred

on January 22 and 24-27, 2016, in the YRMB region.   The air sounding data of Wuhan were used

to compare the structures of the atmospheric boundary layer of the heavy air pollution and clean

air periods. Figure 5 presents the vertical profiles of air temperature, wind velocity and potential

temperature averaged for the heavy $PM_{2.5}$ pollution and clean air periods in January 2016. It can

be clearly seen that the inversion layer of air temperature did not exist during the heavy pollution

periods, but a near-surface inversion layer appeared at the height of about 200 m during the clean

air periods (Fig. 5a). The comparison of vertical profiles of horizontal wind velocity experienced

during the clean air periods further revealed the stronger wind speed observed in the heavy air

pollution period below a height of 850 m located in the atmospheric boundary layer exhibiting the

vertical structure similar to a low-level jet stream (Fig. 5b); these conditions could conduce the

downward mixing of the regionally transported air pollutants and produce a local near-surface

accumulation in the YRMB area with elevated ambient $PM_{2.5}$ concentrations, thus contributing to

a heavy air pollution. To characterize the atmospheric stability in the boundary layer, the vertical

profiles of potential air temperature ($\theta$) were calculated with air temperature and pressure (Fig. 5c).

The vertical change rate of $\theta$ was used to quantify the static stability of the boundary layer in this

study (Oke, 2002;Sheng et al., 2003). A lower vertical change rate of $\theta$ generally indicates a

decreasing stability or increasing instability of the boundary layer. The averaged static stability

values of the near-surface layer below a height of 200 m during the heavy pollution and clean air

periods were approximately $4.4 K \cdot km^{-1}$ and $13.2 K\ km^{-1}$, respectively (Table 3). This obvious



decrease in stability of the boundary layer from clean air to heavy pollution periods reflects an
anomalous tendency for instability in the boundary layer during heavy pollution periods in the
YRMB region during January 2016.
The meteorological conditions of stagnation characterized by weak wind, temperature
inversion and a stable vertical structure of the atmospheric boundary layer is generally accepted as
the typical meteorological drivers for heavy air pollution (An et al., 2019;Ding et al., 2017).
Nevertheless, this study of environmental and meteorological observations in the YRMB region
has revealed a unique meteorological condition of "non-stagnation" in the atmospheric boundary
layer during heavy air pollution periods characterized by strong wind, lack of an inversion layer
and a more unstable structure of the atmospheric boundary layer; these conditions are generally
regarded as the typical pattern of atmospheric circulation that facilitates the regional transport of
air pollutants from upstream source to downwind receptor regions. Regional transport of $PM_{2.5}$
associated with the source-receptor relationship between the air pollution regions in
central-eastern China and the YRMB was investigated based on the observational analysis
described in Sect. 3.1.
**3. Regional transport of $PM_{2.5}$ in heavy air pollution periods**
**3.1 Changes of $PM_{2.5}$ and winds observed in central-eastern China**
The monthly averages of observed $PM_{2.5}$ concentrations and the anomalies of wind speed
averaged in three heavy air pollution periods relatively to the monthly mean wind speed in January
2016 over central-eastern China are shown in Figure 6. In January 2016, a large area of
central-eastern China experienced air pollution with high levels of $PM_{2.5}$ (>75 μg m$^{-3}$), especially



serious in the NCP region and the Fenhe-Weihe Plain in central China (Fig. 6a). As seen in Figure
6, the YRMB region (Site 1, Wuhan) was situated in the downwind southern edge of an observed
air pollution area located over central-eastern China, where the northerly winds of the East Asian
winter monsoon prevail climatologically in January (Ding, 1994). It is notable that the
anomalously stronger northerly winds were observed over the upstream region in central-eastern
China during three periods of wintertime heavy $PM_{2.5}$ pollution in the YRMB region (Fig. 6b).
Driven by the strong northerly winter monsoonal winds (Fig. 6b), the regional transport of air
pollutants from the source regions in central-eastern China could largely contribute to wintertime
heavy air pollution periods in the downwind receptor region of YRMB.

In order to explore the connection of regional transport of $PM_{2.5}$ over central-eastern China

to three events of heavy air pollution in the YRMB region, six observational sites were selected
from the northwestern, northern and northeastern upwind areas located over central-eastern China
(Fig. 6a) to represent the temporal $PM_{2.5}$ and wind variations along the different routes of regional
transport of $PM_{2.5}$ with the southward incursion of stronger northerly winds of East Asian
monsoon across central-eastern China (Fig. 7).    The southeastward movement of heavy $PM_{2.5}$
pollution driven by stronger northerly winds from Luoyang and Xinyang to Wuhan (Sites 3, 2, and
1 in Fig. 6) presents a northwestern route of regional transport of $PM_{2.5}$ for the heavy air pollution
period P1 in the YRMB (see upper panels of Fig. 7). The southwestward advance of $PM_{2.5}$ peaks
governed by winter monsoonal winds the from Tongling and Hefei to Wuhan (Sites 5, 6, and 1 in
Fig. 6) exerted a significant impact on the heavy air pollution period P2 aggravated by regional
transport of $PM_{2.5}$ across Eastern China to the YRMB region (see middle panels of Fig. 7). A
northern pathway of regional transport of $PM_{2.5}$ connected Zhengzhou and Xinyang to Wuhan



(Sites 4, 2, and 1 in Fig. 6) during the YRMB's heavy air pollution period P3 with anomalously
strong northerly winds (see Fig. 6b and lower panels of Fig. 7). It is noteworthy in Fig. 7 that the
heavy $PM_{2.5}$ pollution periods at the upstream sites Hefei, Tongling, Luoyang, Xinyang and
Zhengzhou (Fig. 6a) were generally dispelled by strong northerly winds, while strong northerly
winds could trigger the periods of heavy $PM_{2.5}$ pollution in the YRMB region (Wuhan, Fig. 6), and
such inverse effects of strong winds on heavy air pollution in the source and receptor regions
reflect an important role of regional air pollutant transport in worsening air pollution in the
YRMB's receptor region.
The regional transport over central-eastern China associated with the source-receptor
relationship directing heavy $PM_{2.5}$ pollution to the YRMB region was revealed with observational
analysis.  Backward trajectory modeling was used to further confirm the patterns of regional
transport of $PM_{2.5}$ over central-eastern China and the resulting contribution to heavy air pollution
in the YRMB region, as described in the following Sects.
**3.2 FLEXPART-WRF model**
3.2.1 Model description
The Flexible Particle dispersion (FLEXPART) model (Stohl, 2003) is a Lagrange particle
diffusion model developed by the Norwegian Institute for Air Research (NIAR). In this model, the
trajectory of a large number of particles released from a source is simulated with consideration of
the processes of tracer transport, turbulent diffusion, and wet and dry depositions in the
atmosphere (Brioude et al., 2013). Applying backward trajectory simulation can determine the
distribution of potential source regions that may have an impact on a target point or receptor



region (Seibert and Frank, 2003;Zhai et al., 2016;Chen et al., 2017a; Chen et al., 2017b).
Initially, FLEXPART could be driven by the global reanalysis meteorological data obtained
from the European Centre for Medium-Range Weather Forecasts (ECMWF) or the National
Centers of Environmental Prediction (NCEP). For the refined simulation of air pollutant sources
and transport, FLEXPART was coupled offline with the Weather Research and Forecasting Model
(WRF) to effectively devise the combined model FLEXPART-WRF (Fast and Easter, 2006), which
has been widely used to investigate the potential sources of air pollutants in consideration of
environmental change (Stohl, 2003;De Foy et al., 2011;An et al., 2014;Sauvage et al., 2017).
3.2.2 Model configuration
The WRF model was configured with two nested domains. The coarse domain covered the
entirety of Asia with a 30 km×30 km horizontal resolution, and the nested fine domain included
most of China and surrounding regions with a 10 km×10 km horizontal resolution. The physical
parameterizations used in WRF were selected with the Morrison microphysics scheme (Morrison,
2009), the Rapid Radiative Transfer Model (RRTM) scheme for long and short wave radiation
(Mlawer et al., 1997), the Yonsei University (YSU) boundary layer scheme (Hong, 2006), Grell
3D cumulus parameterization, and the Noah land surface scheme (Grell et al., 2005). Driven with
the reanalysis meteorological data obtained from NCEP for initial and boundary meteorological
conditions, the WRF simulation ran 12 h each time with the first 6 h simulations constituting
spin-up time.
The FLEXPART-WRF simulation was conducted for the 48-hr backward trajectory with a
release of 50,000 $PM_{2.5}$ particles per hour in Wuhan (30.61N, 114.42E) for January 2016. The



48-hr backward trajectory simulation results were output with the residence time of $PM_{2.5}$ particles
in a horizontally resolution of 0.1°×0.1°. The FLEXPART simulations of $PM_{2.5}$ particle residence
time over the 48-hr backward trajectory pathways were multiplied with the regional primary $PM_{2.5}$
emission fluxes to quantify the contribution of regional transport of $PM_{2.5}$ to air quality change in
the YRMB region with identifying the patterns of regional transport of $PM_{2.5}$ over central-eastern
China. The primary $PM_{2.5}$ emission data of 2016 obtained from the Multi-resolution Emission
Inventory for China (MEIC, http://www.meicmodel.org/) were selected for use as the regional
$PM_{2.5}$ emission fluxes in this study.
3.2.3 Validation of modeling results

The simulated meteorology, which included wind speed, air temperature, relative humidity

and surface pressure, were compared with observations at five sites (Wuhan, Changsha, Hefei,
Zhengzhou and Nanchang) over central-eastern China. The correlation coefficients and
normalized standardized deviations were calculated and are shown in Figure 8 (Taylor, 2001).
Based on the results with correlation coefficients passing the significance level of 0.001 and low
normalized standardized deviations (Fig. 8), it was confirmed that WRF-modeled meteorology
that is consistent with observations could be used to drive the FLEXPART backward trajectory
simulation in this study.
**3.3 Contribution of regional transport of $PM_{2.5}$ to heavy pollution**

Based on the FLEXPART-WRF backward trajectory simulation, the upstream sources of

$PM_{2.5}$ emissions for heavy air pollution in Wuhan could be identified. The contribution rates $rate_{i,j}$
of regional transport of $PM_{2.5}$ from the upstream sources to air pollution in the downstream





receptor region of YRMB were calculated by Eq. (1), and the total contribution **R** of regional
transport from the non-local emission sources are estimated by Eq. (2) (Chen et al., 2017b).

$$rate_{i,j} = \frac{E_{i,j} \times r_{i,j}}{\sum_{1,1}^{N,S} E_{i,j} \times r_{i,j}} \qquad (1)$$

$$R = \sum_{(N_1,S_1)}^{(N_2,S_2)} rate_{i,j} \qquad (2)$$

where the subscripts *i* and **j** represent a grid location; $r_{i,j}$ represents the residence time of $PM_{2.5}$
particles simulated by FLEXPART-WRF; and, $E_{i,j}$ represents the $PM_{2.5}$ emission flux over the grid.
The first grid location $(N_1, S_1)$ and the last grid location $(N_2, S_2)$ over the non-local emission
sources and the local area of Wuhan were determined respectively by the regional transport of
$PM_{2.5}$ pathways and the YRMB region as simulated by FLEXPART-WRF.

The non-local emission sources that affected $PM_{2.5}$ concentrations during three heavy

pollution periods through regional transport to the YRMB region were quantified by calculation of
the $PM_{2.5}$ contribution rates with Eq (1). Combining the distribution of high $PM_{2.5}$ contribution
rates with the prevailing winds experienced during the three heavy $PM_{2.5}$ pollution periods, the
spatial distribution of the major pathways of regional transport of $PM_{2.5}$ over central-eastern China
could be recognized as shown in Figure 9. During the heavy air pollution period P1 in the YRMB
region, the regional transport of air pollutants was centered along a northwestern route from the
Fenhe-Weihe Plain in central China and a northeastern route from the YRD region (Fig. 9a). The
YRD emission sources of air pollutants in East China exerted an important impact on the heavy air
pollution period P2 through regional transport of $PM_{2.5}$ cross East China to the YRMB region
along the north side of Yangtze River (Fig. 9b). Two major regional transport pathways of $PM_{2.5}$
indicated by the spatial distribution of high contribution rates of $PM_{2.5}$ from the NCP and YRD



regions respectively to the elevated PM$_{2.5}$ concentrations during the YRMB's heavy air pollution
period P3 (Fig. 9c). Governed by the northerly winds of the East Asian winter monsoon, the
regional transport of air pollutants from the central-eastern air pollutant emission source regions in
China provided a significant contribution to the wintertime heavy PM$_{2.5}$ pollution observed in the
YRMB region (Figs. 6-7), which was confirmed by the results of the FLEXPART-WRF backward
trajectory simulation utilized in this study.
In this study, the PM$_{2.5}$ contributions of regional transport to air pollution in the downwind
receptor region could be approximately estimated based on the product of the residence time of
PM$_{2.5}$ particles during regional transport simulated by FLEXPART-WRF, and the PM$_{2.5}$ emission
flux over the source grid. The PM$_{2.5}$ contributions of regional transport over central-eastern China
to PM$_{2.5}$ concentrations during three heavy PM$_{2.5}$ pollution periods P1, P2 and P3 in the YRMB
region were estimated using Eq. (2) with resulting high contribution rates of 68.1%, 60.9%and
65.3%, respectively (Table 4), revealing the significant contribution of regional transport of PM$_{2.5}$
over central-eastern China to the enhancement of PM$_{2.5}$ levels in the YRMB area during
wintertime heavy air pollution periods.
Normally people rely on 3-D numerical models with process analysis capability such as
integrated process rates (IPRs) to quantify the contributions of regional transport to the occurrence
of air pollution episodes. It should be pointed out that the simulations with a Lagrange particle
dispersion model FLEXPART-WRF are utilized to calculate the percentage contribution of
regional transport with identifying the transport pathway in this study.   The major uncertainty of
this method for such calculation as compared to other methods like IPRs is that    the physical and
chemical processes such as wet-deposition and chemical conversion for the formation of



secondary particles are not introduced in the FLEXPART-WRF simulation, which could represent
the basic features of contribution and patterns of regional transport of $PM_{2.5}$ over central-eastern
China when limited to the primary $PM_{2.5}$ particles highlighted in this study.

## 4. Conclusions

This study investigated the ambient $PM_{2.5}$ variations over Wuhan, a typical urban YRMB

region in central-eastern China in January 2016 through analysis of observational data of
environment and meteorology, as well as via FLEXPART-WRF simulation to explore 1) the
meteorological processes involved in the regional transport of air pollutants and 2) regional
transport patterns of $PM_{2.5}$ with the contribution to the air pollution in the YRMB region. Based on
observation and simulation studies on the meteorological conditions of air pollution events in
January 2016 and    regional transport of $PM_{2.5}$ to heavy air pollution over the YRMB region, it is
revealed heavy air pollution with the unique "non-stagnant" atmospheric boundary layer in the
YRMB region aggravated by regional    transport of $PM_{2.5}$ over central and eastern China.

The study of the effects of meteorology and regional transport of $PM_{2.5}$ on    heavy air

pollution were focused on three heavy $PM_{2.5}$ pollution periods in January 2016. The heavy
pollution episodes observed with the peak of $PM_{2.5}$ concentrations exceeding $471 \mu g \ m^{-3}$ over the
YRMB region were characterized by a short duration of less than 26 hr from rapid outbreak to fast
dissipation.

The "stagnation" meteorological condition in the boundary layer characterized by weak wind,

air temperature inversion and a stable vertical structure of the atmospheric boundary layer is
currently accepted as a typical meteorological driver for heavy air pollution. Conversely, this study



of environmental and meteorological observations in the YRMB region revealed a unique
"non-stagnation" meteorological condition of the boundary layer characterized by strong wind, no
inversion layer and a more unstable structure in the atmospheric boundary layer associated with
heavy air pollution periods with excessive $PM_{2.5}$ concentrations in the YRMB region, which
facilitates understanding of the air pollutant source-receptor relationship of regional air pollutant
transport.
Although the emissions and local accumulation of air pollutants in the YRMB could lead to
the formation of light air pollution, in regards to $PM_{2.5}$, over the YRMB region, the regional
transport of $PM_{2.5}$ from central-eastern emission source regions in China contributed significantly
to 65% of the exceedances of $PM_{2.5}$ concentrations during wintertime heavy air pollution periods
in the downwind YRMB region in January 2016, as governed by the strong northerly winds of the
East Asian winter monsoon.
Based on the variations of air quality and meteorology in a typical urban YRMB region in
January 2016, this study revealed a unique "non-stagnant" meteorological condition for the
development of heavy air pollution in the YRMB region with strong contributions of regional
transport of $PM_{2.5}$ over central-eastern China. These conditions and contributions can be
investigated further with climate analyses of long-term observations and a more comprehensive
modeling of air quality and meteorology.
**Data availability:** Data used in this paper can be provided by Chao Yu (ychao012@foxmail.com)
upon request.

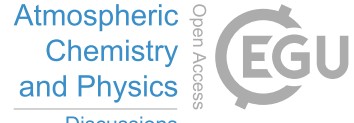

**Author contributions:** CY, TZ and YB conducted the study design. XY, LZ and SK provided the

observational data. LZ assisted with data processing. CY wrote the manuscript with the help of TZ

and XY. YB, SK, JH, CC, YY, GM, MW and JC were involved in the scientific interpretation and

discussion. All of the authors provided commentary on the paper.

**Competing interests:** The authors declare that they have no conflicts of interest.

**Acknowledgement:** This study was jointly funded by the National Natural Science Foundation of

China (41830965; 91744209), the National Key R & D Program Pilot Projects of China

(2016YFC0203304) and the Postgraduate Research & Practice Innovation Program of Jiangsu

Province (KYCX18_1027).

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

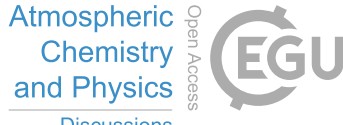




**Table 1.** Correlation coefficients between hourly $PM_{2.5}$ concentrations and meteorological
elements over Wuhan in January 2016.

| Correlation coefficient | WS | T | P | RH |
|---|---|---|---|---|
| $PM_{2.5}$ | 0.10 | 0.31 | -0.47 | 0.20 |


**Table 2.** Correlation coefficients of $PM_{2.5}$ concentrations with wind speed and air temperature in
different air quality levels during the study period.

| Air quality | $PM_{2.5}$ levels | Number of samples | WS | T |
|---|---|---|---|---|
| Clean | $PM_{2.5}<75\mu g\cdot m^{-3}$ | 73 | -0.20 | 0.56 |
| Light pollution | $75\mu g\cdot m^{-3}\leq PM_{2.5}<150\mu g\cdot m^{-3}$ | 135 | -0.19 | 0.15 |
| Heavy pollution | $PM_{2.5}\geq150\mu g\cdot m^{-3}$ | 37 | 0.41 | -0.08 |






**Table 3.** Atmospheric static stability below heights of 200 m in the boundary layer during heavy
pollution and clean air periods with the anomalies relative to the average over January, 2016 in
Wuhan.

| Period | heavy pollution period | clean air period | monthly average |
|---|---|---|---|
| | $(K \cdot km^{-1})$ | $(K \cdot km^{-1})$ | $(K \cdot km^{-1})$ |
| Static stability | 4.4 | 13.2 | 8.6 |
| Anomalies of stability | -4.2 | 4.6 | - |


**Table 4.** The relative contributions of regional transport over central-eastern China to three $PM_{2.5}$
heavy pollution periods P1, P2 and P3 in the YRMB with the local contributions.

| Contribution rates | P1 | P2 | P3 | Averages |
|---|---|---|---|---|
| Regional transport | 68.1% | 60.9% | 65.3% | 65.1% |
| Local contribution | 31.9% | 39.1% | 34.7% | 34.9% |




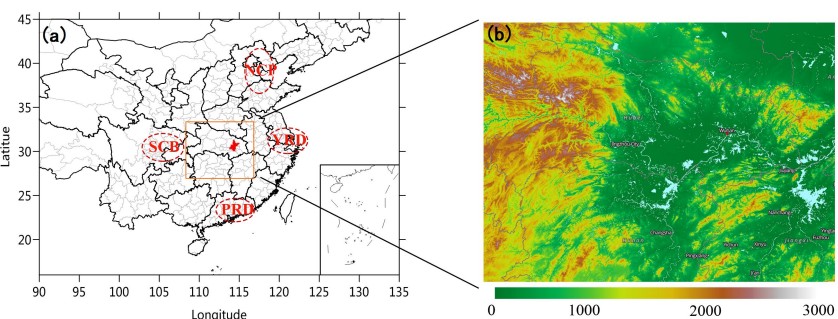


**Fig. 1.** (a) Distribution of the Yangtze River Middle Basin (orange rectangle) with the location of

Wuhan (red area) and the major haze pollution regions of NCP, YPD and SCB in central-eastern

China as well as (b) the YRMB region with terrain height (color contours, m in a.s.l.), the rivers

and lake network (blue areas),downloaded from https://worldview.earthdata.nasa.gov.


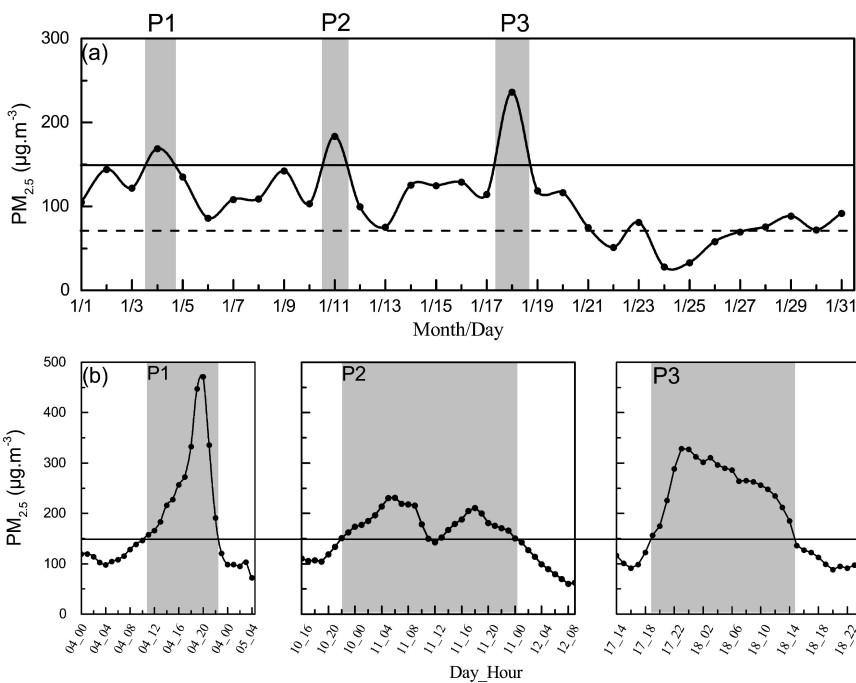




**Fig. 2.** (a) The daily changes of surface PM$_{2.5}$ concentrations in Wuhan in January 2016 with
PM$_{2.5}$ concentrations exceeding 75 μg·m$^{-3}$ (dash line) and 150 μg·m$^{-3}$ (solid lines), respectively,
for light and heavy haze pollution, and (b) the hourly variation of surface PM$_{2.5}$ concentrations in
three heavy air pollution events P1, P2 and P3 with excessive PM$_{2.5}$ levels (>150 μg m$^{-3}$) marked
by the shaded areas.

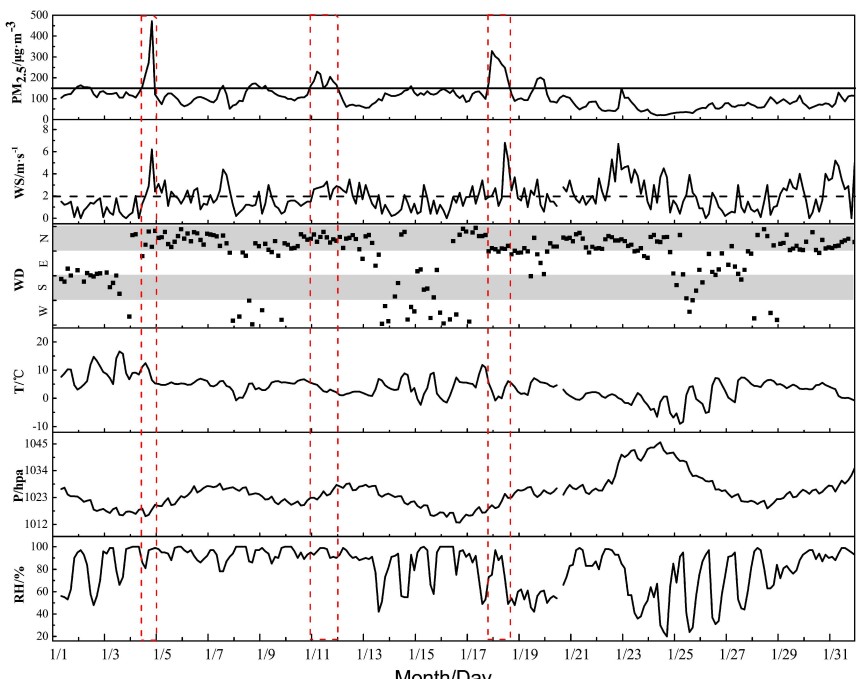


**Fig. 3.** Hourly variations of meteorological elements and PM$_{2.5}$ concentrations in Wuhan in
January 2016 with heavy air pollution periods marked with the columns in red dash lines and
PM$_{2.5}$ concentrations exceeding 150 μg·m$^{-3}$ (solid line).





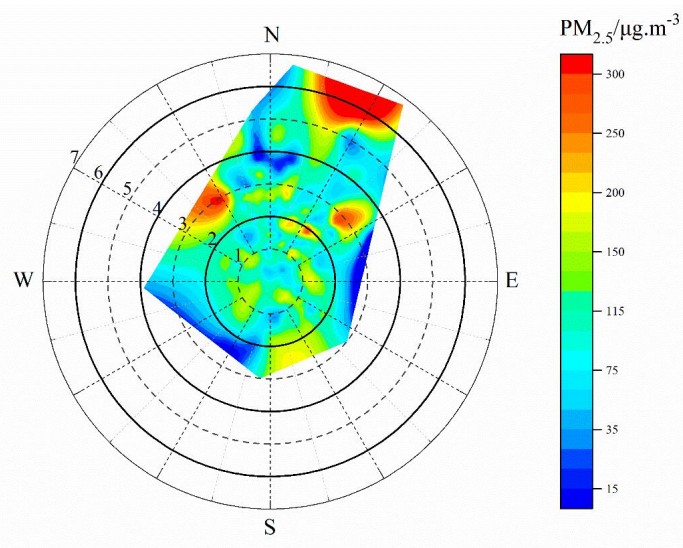


**Fig. 4.** A polar plot of hourly variations in wind speed (round radius, units is m·s⁻¹) and direction
(angles) to surface PM$_{2.5}$ concentrations (color contours, units is μg·m⁻³) in Wuhan in January,

661 2016.

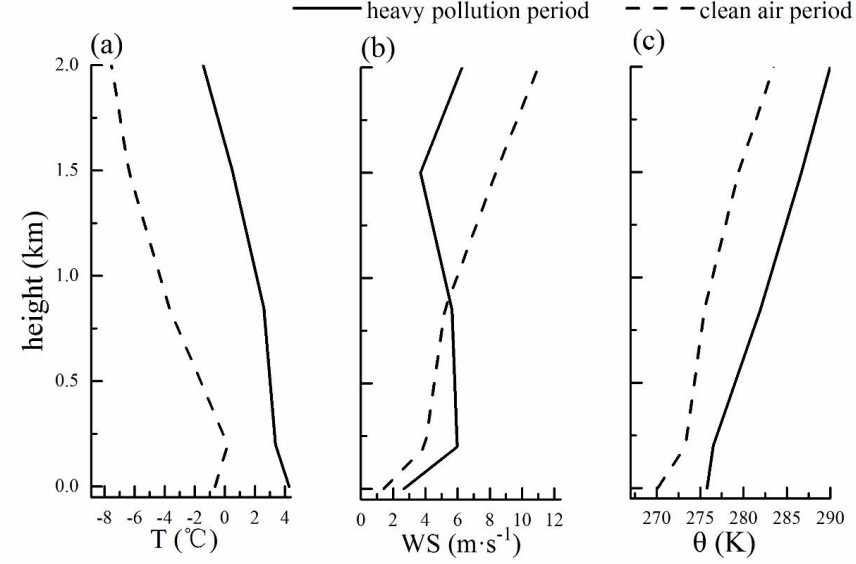


**Fig. 5.** Vertical profiles of (a) air temperature, (b) wind velocity and (c) potential temperature



averaged in heavy PM2.5 pollution and clean air periods over Wuhan during January 2016.

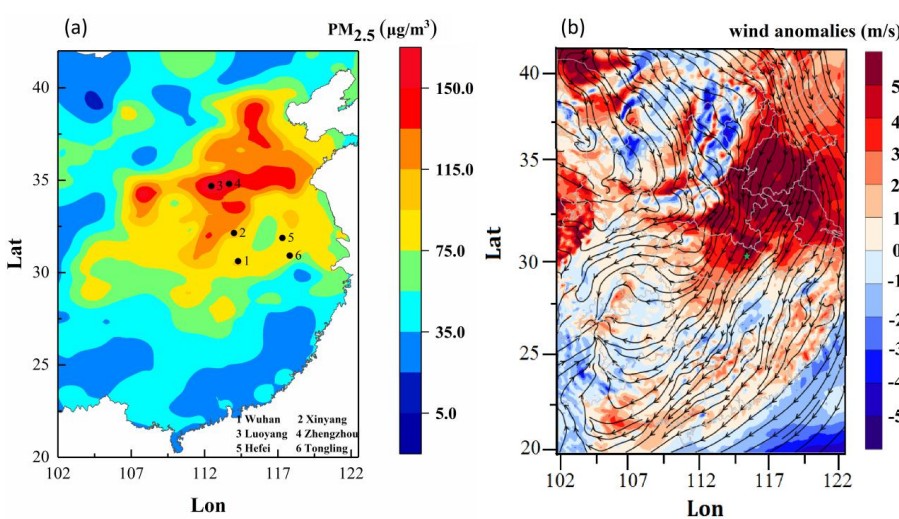


**Fig. 6** Distribution of (a) monthly averages of surface PM2.5 concentrations observed in January
2016 over central-eastern regions in mainland China with the locations of six sites 1. Wuhan, 2.
Xinyang, 3. Luoyang, 4. Zhengzhou, 5. Hefei and 6.Tongling as well as (b) the anomalies (color
contours) of 200m wind speeds averaged during three heavy air pollution periods relatively to the
monthly wind averages (streamlines) in January 2016 over central-eastern China with the location
of Wuhan (a light blue star).



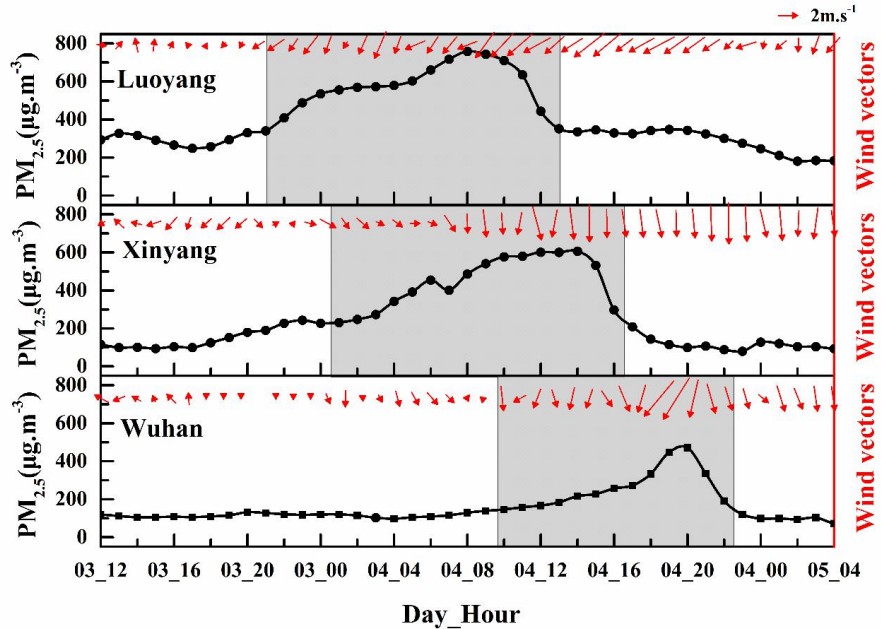






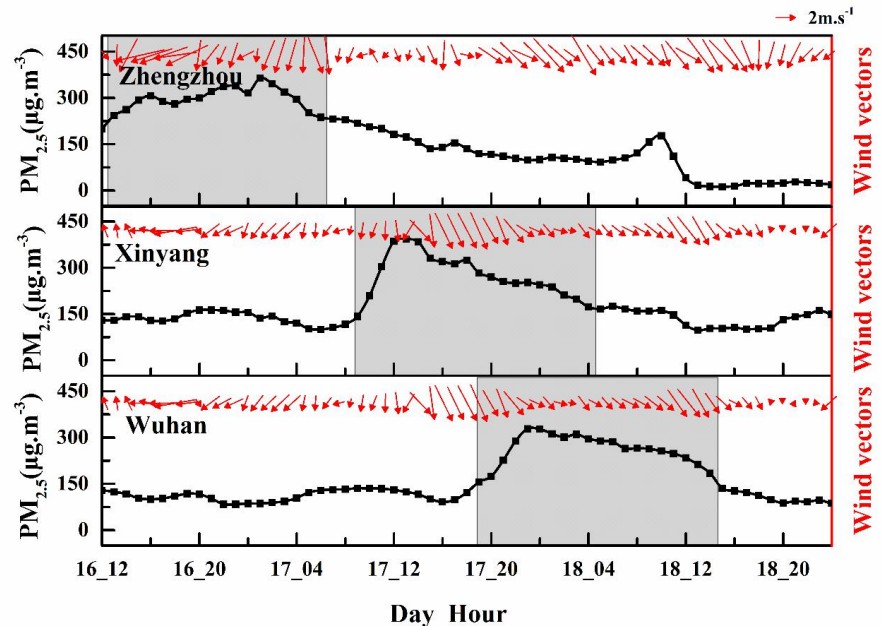


**Fig. 7.** Temporal changes of PM$_{2.5}$ concentrations (dot lines) and near-surface winds (vectors)

observed at five upstream sites (Fig. 6) and Wuhan with shifts of PM$_{2.5}$ peaks (marked with shaded
areas) to the YRMB's heavy PM$_{2.5}$ pollution periods P1 P2 and P3 (respectively in upper, middle
and lower panels) in January 2016.





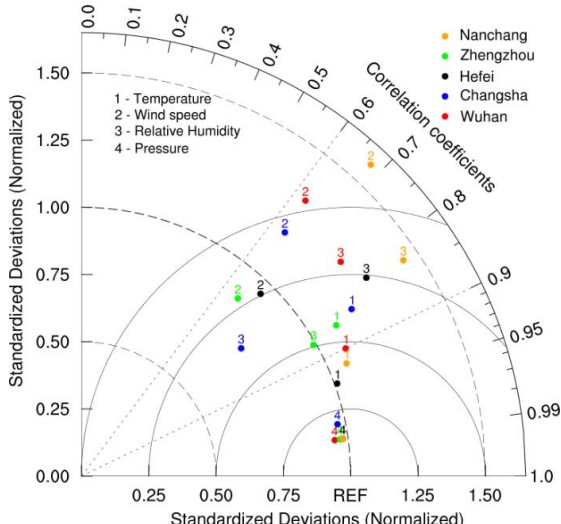


**Fig. 8.** Taylor plots with the normalized standard deviations and correlation coefficients between
simulated and observed meteorological fields. The radian of the sector represents the correlation
coefficient, the solid line indicates the ratio of standard deviation between simulations and
observations, the distance from the marker to "REF" reflect the normalized root-mean-square error
(NRMSE).

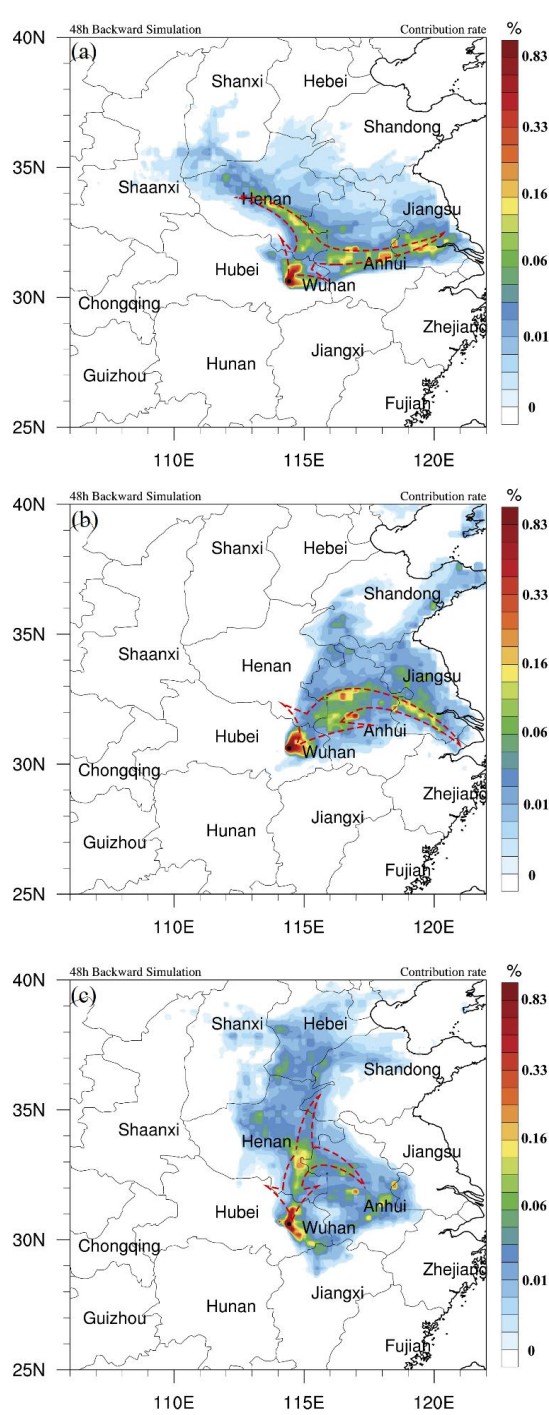


**Fig. 9.** Spatial distribution of contribution rates (color contours) to PM$_{2.5}$ concentrations in Wuhan




with the major pathways of regional transport over central-eastern China (dash arrows) for (a)
heavy pollution periods P1, (b) P2 and (c) P3 in January, 2016 simulated by the model
FLEXPART-WRF.