# Peer review of "Heavy air pollution with the unique “non-stagnant”"

_Atmospheric Chemistry and Physics, 2019_

## Referee Comment (RC1) · Anonymous Referee #1 · 25 Dec 2019

Review of Yu et al "Heavy air pollution with the unique "non-stagnant" atmospheric boundary layer in the Yangtze River Middle Basin aggravated by regional transport of $PM_{2.5}$ over China"

Yu et al investigated the impacts of regional transport to the heavy haze pollution in January 2016 over Wuhan, a city located over the Yangtze River Middle Basin in the central part of China. This study characterized unique "non-stagnant" conditions (e.g., high winds, no inversion layers) associated with extreme high levels of $PM_{2.5}$ concentrations (e.g., strong correlation between $PM_{2.5}$ concentrations greater 150 μg m$^{-3}$ and wind speed), which differed significantly from traditional haze pollutions with low near-surface winds and inversion layers found in the literatures. The authors employed both observational and modeling analyses to prove the importance of the contribution of regional transport to the excessive $PM_{2.5}$ concentrations over Wuhan. This is an interesting study to demonstrate the complexity and challenge of the severe haze pollution over central-eastern China during wintertime, with research scope aligned with topics suitable for ACP. However, the current format of the manuscript is not accepted, due to ambiguous structure of the manuscript, lack of detailed descriptions of observational and modeling methods, concerns of technical methodology as well as numerous grammar errors and typos over the entire manuscript. A major revision is needed for this manuscript before further consideration of publication in ACP. My comments for the manuscript are shown as follows.

**Major Comments**

1. Research Methodology and Results/Discussions for the paper are not clear

I have difficulty in following the paper's research methodology/results. The authors mix the research methodology and results in the same section. I highly recommend

that the authors should re-organize the structure of the paper. The descriptions of observational data from various sites and FLEXTPART-WRF (Sect. 3.2.1 and Sect. 3.2.2) should be placed in Section 2. And Results and Discussions, including the analysis of the observational data and modeling study, should be placed in Section 3.

2.  The descriptions of the data used in this study are not adequate and needed to be expanded to provide a more detailed and rigorous documentation.

We don't know the spatial locations of the observational sites for $PM_{2.5}$ measurements, especially ten sites over Wuhan, which need to be presented. A spatial map of WRF modeling domain, with $PM_{2.5}$ measurement sites inserted, will be very helpful. Moreover, what is the measurement technique used for $PM_{2.5}$? What is the measured frequency/quality data control method, and measurement uncertainty associated with $PM_{2.5}$ concentrations and other meteorological parameters for each site? How do you represent Wuhan's hourly $PM_{2.5}$ concentrations out of the ten measurement sites? And how do you calculate the correlation coefficients between $PM_{2.5}$ concentrations and wind speed/temperature over Wuhan in January 2016 out of ten measured sites?

3.  In terms of quantification of regional transport contributions for $PM_{2.5}$ over Wuhan, the authors have utilized FLEXPART-WRF model. However, I have concerns about the convolution of FLEXPART-WRF residence time with the $PM_{2.5}$ bottom-up emission fluxes from MEIC. Firstly, what is the definition of residence time here? Is it the $PM_{2.5}$ lifetime? With Lagrangian method, it will result in a Jacobian matrix (footprint), in unit of mass per volume per unit flux. It is helpful for the authors to mathematically derive the residence time for particles out of FLEXPART, the product of the residence time and the bottom-up emission flux, and ultimately the regional transport contribution rate in the "Research Methodology" Section. The authors should insert the unit for each variable out of FLEXPART modeling. Meanwhile, please help the readers about the purpose of the WRF model here. Further, FLEXPART does not

consider chemistry and deposition in the model, the only part it accounts for is the transport, driven by reanalysis data. $PM_{2.5}$ contains a significant portion of secondary organic and inorganic aerosols, which come from important and complex physiochemical processes in the atmosphere. How this methodology (FLEXPART-WRF) is proven robustness to quantify the regional transport contribution? What is the uncertainty range here?

**Minor Comments**

Line 48: The order of the references is messed up, which should follow the order of the first letter of the first author for each reference alphabetically, and should be "An et al., 2019; Fuzzi et al., 2015; Nel, 2005" for this case. Please check the entire manuscript.

Line 50: The definition of $PM_{2.5}$ "particulate matter with an aerodynamical diameter equal to or less than 2.5 micrometers".

Line 99: change "humid environment. (see Fig. 1b)" to "humid environment (see Fig. 1b)". There are so many similar typos across the entire manuscript. Please CHECK!

Line 101: The associated temporal variations of $PM_{2.5}$ concentrations for the study period out of ten sites in Wuhan are strongly recommended to be plotted and placed in the Supplemental.

Line 107: Change "obviously" to "obvious".

Line 124: "heavy $PM_{2.5}$ pollution the over central-eastern China" should be revised as "heavy $PM_{2.5}$ pollution over the central-eastern China".

Line 128: The number and unit should be separated (75 $\mu g\ m^{-3}$). Similar changes should be applied for the entire manuscript.

Line 146: "at same day." should be changed to 'at the same day,".

Lines 147-Line 149: The authors use "am" and "a.m." interchangeable. Please be consistent for the entire manuscript. Similar for "pm" and "p.m.".

Lines 161-165: Grammar error here. Please re-write this sentence. And what is the logical relationship between this sentence and the previous one? Do you try to demonstrate the reasons for this result? If so, probably it is better to begin the sentence with "There are several reasons associated with this result. Firstly, ......".

Line 165: what is "CEC" here?

Lines 165-170: There are many typos and grammar errors in this sentence. And I am confused by this sentence as well, which looks very odd to me. Is this your statement or conclusion? Several references to support your statement will be necessary.

Lines 184-185: There should be spaces between references, which should be "(Miao et al., 2018; Xu et al., 2016b). There are many cases (e.g., Line 187, 254, 263 and etc) like this. Please check over the entire manuscript.

Line 210: "the stagnation meteorological conditions" should be revised as "meteorological conditions of the stagnation".

Lines 233-234: References relevant to secondary organic and inorganic aerosols study over Wuhan?

Line 276: Change "relatively" to "relative".

Lines 296-299: First of all, there are grammar errors in this sentence (e.g., ...by winter monsoonal winds the from Tongling and Hefei to Wuhan (...). Second of all, the site numbers of Tongling and Hefei are 6 and 5 respectively, as indicated by legend of Figure 6a?

Lines 311-313: It seems that this sentence belongs to the beginning of Section 3.2.

Lines 331-333: I recommend that the authors make a plot associated with the modeling domains, which demonstrates the regions with the coarse and finer horizontal resolutions (refer to my major comment #2).

Lines 341-342: I have concerns about the release of the number of particles in FLEXPART-WRF. Firstly, for particles from FLEXPART, it is not PM2.5 particles, it is just particles to represent the air parcels. Secondly, can you double check that the model release 50,000 particles per hour? From my understanding, for each hourly mean $PM_{2.5}$ observation at the receptor list, the release of particles in the 48-h backward trajectory simulation in FLEXTPART just happens in the first hour, with the rest of the time tracking the routes/transport of the particles over the simulation domain?

Line 374: Change "Eq (1)" to "Eq. (1)".

Lines 634-637: For "K km-1", it should be "K km$^{-1}$".

Lines 640-645: There are many typos for Figure 1. For Y-axis title in Figure 1a, it should be "Latitude". Moreover, both units of X-axis and Y-axis in Figure 1a are missing. In Line 643, "YPD" is a typo. And where is the description of PRD here?

Lines 663-664: The solid line for heavy $PM_{2.5}$ pollution and the dash line for clean air period are missing in the caption for Figure 5.

Lines 679-680: Why there are no "comma" among "P1 P2 and P3". I suggest changing the caption of the last part of the caption of Figure 7 as "....pollution periods of P1 (upper panel), P2 (middle panel) and P3 (lower panel), respectively, in January 2016".

---

## Referee Comment (RC2) · Anonymous Referee #2 · 31 Dec 2019

General comments.

In this manuscript, the authors present an observational analysis to characterize the unique features of meteorological conditions that account for the heavy air pollution events in Wu Han, a metropolis in the Yangtze River Middle Basin, China, and then use a Lagrange particle dispersion model to quantify the percentage contribution of regional transport to such heavy pollution events. They found that PM$_{2.5}$ concentrations show a positive correlation with wind speeds and no stable atmospheric boundary conditions are required to support the accumulation of air pollutants when 24-hr average PM$_{2.5}$ concentrations are higher than $150.0\ \mu g \cdot m^{-3}$. Regional transport driven by strong wind speed contributed more than 65% increase in surface PM$_{2.5}$ concentrations during the development of air pollution events in this region. The study represents a great interest to air quality community given the unique features which are very different from those presented in the textbooks. This version is improved to some extent as compared to the first submission. Part of my comments have been addressed but not all. Especially, the manuscript structure is not re-organized as suggested, a lot of grammar errors or typos need to be corrected throughout the manuscript. In addition, I have several major concerns with the authors' arguments during their analyses and discussion. Thus, a major revision is still required before it is accepted for publication.

Major comments
1. It is strongly recommended to re-organize the structures of the manuscript. Both Methodology and Results/Discussion parts are mixed together in the current version. So it is suggested to move "Model Description (Section 3.2.1)", "Model Configuration (Section 3.2.2)", and the way of calculating "contribution rates" (Lines 360-375 in Section 3.3) to a new section like "Data and methods"(say Section 2 in the new version), and then move part of current Section 2.1, Sections 2.2 and 2.3 to Section 3 like "Results and Discussion" in the revised or new version something like that.

2. The East Asian winter Monsoon were mentioned at least 10 times throughout the manuscript to highlight its importance in driving the regional transport during development of heavy pollution events observed in Wuhan. As we know, the East Asian Monsoon represents a seasonal mean behavior and its temporal scale is much longer than that of air pollution events which usually have a scale of one to several day(s) but not longer than one week according to the authors' argument. The authors need show some scientific evidences to support their arguments on how the East Asian winter Monsoon can drive the regional transport which may lead to the development of heavy pollution events. Otherwise, the readers may get confused when they read Fig.9b in which the regional transport was from East China other than North China. My suggestion is to limit the emphasis of the East Asian winter Monsoon in this study.

3.  Estimate of percentage contribution of regional transport to the heavy pollution events in the YRMB region is one of the major works proposed by this study. As described in Eqs. 1 and 2, simulation of residence time of $PM_{2.5}$ is critical to conduct such calculations. Please define residence time. How does the FLEXPART simulate the residence time?  A little bit more details are helpful for our readers to understand the percentage contribution of regional transports to the three different episodes.

4.  Lines 323-329:  I assume that the FLEXPART simulations were driven by the WRF outputs rather than ECMWF or NCEP reanalysis data. If this is the case, please make clarification and delete lines 323-325.

5.  Fig.5b:  We can see that the heavy air pollution events had stronger winds within the 1-km layer but weaker winds above the 1-km layer as compared to that light air pollution events. Does this mean that regional transport is mainly limited to the 1-km layer? Some discussions on this will be helpful.

6.  Writing needs a heavy edit work.  There are a lot of grammar errors or typos and many sentences need further improvement. Some of examples include "obviously differences (L107)", "relative high (L109)", "suffering under significant (L133)", "has significantly influence (L162)", "relatively to (L276)", "a horizontally resolution (L344)", etc. I am not going to list all of them since there are many.

Minor comments:

1.  L19:  central China→ Central China.
2.  L20: I am not sure "excessive" is appropriate in this manuscript.
3.  L30: I did check "List of regions of China" at Wikipedia at https://en.wikipedia.org/wiki/List_of_regions_of_China, and didn't find "central-eastern China". So "Central China" should be better and sufficient.
4.  L33: FLEXPART-WRF or WRF-FLEAPART? I would suggest the latter since it is WRF-driven FLEXPART. In addition, please define any abbreviated terms at its first appearance. Please check similar issue for other abbreviations throughout the manuscript.
5.  L155-157: Please define these abbreviations at their first appearances.
6.  L251: change "the atmospheric stability in the boundary layer" to "the stability of the atmospheric boundary layer"?
7.  L261-262:  Please change "is generally accepted" to "are generally accepted".
8.  L272: Are you sure "it is in Section 3.1"?
9.  L287: Please add "the" before "YRMB".
10. L342:  Please change to (30.61$^{o}$N, 114.42$^{o}$E).
11. L232-235: I feel a "jump" when I read this sentence.
12. L352: "The simulated meteorology" → "The simulated meteorological fields".
13. L373-374:  Change "by calculation of the $PM_{2.5}$ contribution rates with Eq (1)" to "by using the $PM_{2.5}$ contribution rates calculated with Eq.1" something like that.

14. L309-313: I do not think this paragraph is necessary since it does not provide any useful information. Similar issue can be found in other places of the manuscript.

15. L338-339: What are the horizontal resolutions of the NCEP reanalysis data?

16. L419-423: Does this paragraph represent any significant findings or conclusions obtained from this study?  I am not sure including this paragraph is necessary here.

17. L424-426: We know this already and I don't think you need iterate this sentence here. It does not provide any more useful information.

18. L625-629: Please define WS, T, P, and RH in the description of Table 1 and Table 2.

19. Fig.1b: The font size of those cities shown in Fig.1b is too small. Is it possible to add the locations of 10 sites presented on Page 5 at Lines 101-103 in this plot?

20.  Fig.9: I believe that the values of the percentage contribution rates are not correct.

---

## Author Comment (AC1) · 21 Jan 2020

Dear Editors and Referees:

Thank you very much for your constructive suggestions and helpful comments for improving our manuscript acp-2019-758. We have accordingly made the careful revisions. Revised portions are highlighted in the revised manuscript. In the following we quoted each review question in the square brackets and added our response after each paragraph.

**Responses to Referee #1**

*[Yu et al investigated the impacts of regional transport to the heavy haze pollution in January 2016 over Wuhan, a city located over the Yangtze River Middle Basin in the central part of China. This study characterized unique "non-stagnant" conditions (e.g., high winds, no inversion layers) associated with extreme high levels of $PM_{2.5}$ concentrations (e.g., strong correlation between $PM_{2.5}$ concentrations greater 150 μg $m^{-3}$ and wind speed), which differed significantly from traditional haze pollutions with low near-surface winds and inversion layers found in the literatures. The authors employed both observational and modeling analyses to prove the importance of the contribution of regional transport to the excessive $PM_{2.5}$ concentrations over Wuhan. This is an interesting study to demonstrate the complexity and challenge of the severe haze pollution over central-eastern China during wintertime, with research scope aligned with topics suitable for ACP. However, the current format of the manuscript is not accepted, due to ambiguous structure of the manuscript, lack of detailed descriptions of observational and modeling methods, concerns of technical methodology as well as numerous grammar errors and typos over the entire manuscript. A major revision is needed for this manuscript before further consideration of publication in ACP. My comments for the manuscript are shown as follows.]*

**Response 1:** Many thanks for the encouraging comments and constructive suggestions on our manuscript acp-2019-758. Accordingly, we have restructured the manuscript with detailed descriptions of observational and modeling methods, concerns of technical methodology as

well as corrected the grammar errors and typos over the entire manuscript (please find them in the following responses and the highlighted revisions in the revised manuscript).

*[Major Comments*

*1. Research Methodology and Results/Discussions for the paper are not clear I have difficulty in following the paper's research methodology/results. The authors mix the research methodology and results in the same section. I highly recommend that the authors should re-organize the structure of the paper. The descriptions of observational data from various sites and FLEXTPART-WRF (Sect. 3.2.1 and Sect. 3.2.2) should be placed in Section 2. And Results and Discussions, including the analysis of the observational data and modeling study, should be placed in Section 3.]*

**Response 2:** Following the referee's suggestions, we have re-organized the structure of the paper. In the revised manuscript, the descriptions of observational data from various sites and FLEXTPART-WRF (Sect. 3.2.1 and Sect. 3.2.2) are placed in Section 2. And Results and Discussions, including the analysis of the observational data and modeling study, are placed in Section 3.

*[2. The descriptions of the data used in this study are not adequate and needed to be expanded to provide a more detailed and rigorous documentation.*
*We don't know the spatial locations of the observational sites for $PM_{2.5}$ measurements, especially ten sites over Wuhan, which need to be presented. A spatial map of WRF modeling domain, with $PM_{2.5}$ measurement sites inserted, will be very helpful. Moreover, what is the measurement technique used for $PM_{2.5}$? What is the measured frequency/quality data control method, and measurement uncertainty associated with $PM_{2.5}$ concentrations and other meteorological parameters for each site? How do you represent Wuhan's hourly $PM_{2.5}$ concentrations out of the ten measurement sites? And how do you calculate the correlation coefficients between $PM_{2.5}$ concentrations and wind speed/temperature over Wuhan in January 2016 out of ten measured sites?]*

**Response 3:** According to the referee's suggestions, we have added the spatial locations of the ten observational sites for PM$_{2.5}$ measurements over Wuhan in the supplemental (Fig. s1). Besides, the PM$_{2.5}$ data used in this study were collected from the national air quality monitoring network operated by the Ministry of ecology and environmental protection of China, The mass volume concentrations of surface PM$_{2.5}$ are operationally hourly observed with the instrument of the Thermo Fisher Scientifi. The observation data are under quality control based on the China's national standard of air quality observation before released by the Ministry of ecology and environmental protection of China. The source of the data has been added in the revised manuscript. At last, the surface PM$_{2.5}$ concentrations averaged over 10 observation sites in Wuhan are used to calculate the correlation coefficients with the changing meteorological drivers (wind speed/temperature etc.) over Wuhan in January 2016 to investigate the local meteorological influences on hourly changes of surface PM$_{2.5}$ concentrations in Wuhan.

*[3. In terms of quantification of regional transport contributions for PM2.5 over Wuhan, the authors have utilized FLEXPART-WRF model. However, I have concerns about the convolution of FLEXPART-WRF residence time with the PM2.5 bottom-up emission fluxes from MEIC. Firstly, what is the definition of residence time here? Is it the PM2.5 lifetime? With Lagrangian method, it will result in a Jacobian matrix (footprint), in unit of mass per volume per unit flux. It is helpful for the authors to mathematically derive the residence time for particles out of FLEXPART, the product of the residence time and the bottom-up emission flux, and ultimately the regional transport contribution rate in the "Research Methodology" Section. The authors should insert the unit for each variable out of FLEXPART modeling. Meanwhile, please help the readers about the purpose of the WRF model here. Further, FLEXPART does not consider chemistry and deposition in the model, the only part it accounts for is the transport, driven by reanalysis data. PM2.5 contains a significant portion of secondary organic and inorganic aerosols, which come from important and complex physiochemical processes in the atmosphere. How this methodology (FLEXPART-WRF) is proven robustness to quantify the regional transport contribution? What is the uncertainty range here?]*

**Response 4:** Thanks for the comments. In the revised manuscript, we have clarified the quantification of regional transport contributions with utilizing the model FLEXPART-WRF in the revised manuscript as followings:

In the model FLEXPART-WRF, the trajectory of a large number of particles released from a source is simulated with consideration of the processes of tracer transport, turbulent diffusion, wet and dry depositions in the atmosphere. With Lagrangian method, it could result in a Jacobian matrix (footprint), in unit of mass per volume per unit flux. Stohl et al. (2005) mathematically derived the residence time for particles out of FLEXPART. Generally, in the backward trajectory of FLEXPART modeling, a large number of particles is released at a receptor and transported backward in time. Then the residence time (not the lifetime) of all particles, normalized by the total number of released particles, is determined on a uniform grid. In this study for the receptor of Wuhan, the residence time for a thickness of 100 m above the surface was calculated and considered the ''footprint'' (in unit of s). By multiplying the residence time with the air pollutant emission flux in the respective grid cell (in unit of μg $m^{-2}$ $s^{-1}$ ) calculated from the Multi-resolution Emission Inventory of year 2016 for China (MEIC, http://www.meicmodel.org/), the emission source contribution (in μg $m^{-2}$) from this grid cell to the receptor could be estimated (Stohl, 2003; Stohl et al., 2005;Ding er al.,2009), yielding a so-called potential source contribution map, which is the geographical distribution of the regional transport contribution rates (%) of the emission source grid cell to $PM_{2.5}$ pollution at the receptor of Wuhan (Fig. 9).

A need for further multiscale modeling and analysis has encouraged new developments in FLEXPART-WRF, a FLEXPART version that works with the Weather

Research and Forecasting (WRF) mesoscale meteorological model (Brioude et. al., 2013). For the refined simulation of air pollutant sources and transport, FLEXPART modeling driven by mesoscale meteorology from WRF modeling has been widely used to investigate the potential sources of air pollutants in consideration of air pollution change.

In this study, the $PM_{2.5}$ contributions of regional transport to air pollution in the downwind receptor region could be approximately estimated based on the product of the residence time of air particles during regional transport simulated by FLEXPART-WRF, and the $PM_{2.5}$ emission flux over the source grid in Central and Eastern China. The potential source contribution is estimated based on transport alone, ignoring chemical and removal processes. We also understand that the physical and chemical processes such as complex deposition and chemical conversion for the formation of secondary particles are not introduced in the FLEXPART-WRF emulation, which could represent the basic features of contribution and patterns of regional $PM_{2.5}$ transport over central and eastern China, when limited to the primary $PM_{2.5}$ particles highlighted in this study. Considering less precipitation in the winter monsoon season, how this methodology (FLEXPART-WRF) is proven robustness to quantify the regional transport contribution with the uncertainty range here could mostly rely on a portion of secondary organic and inorganic aerosols, which are resulted from important and complex physiochemical processes in the atmosphere.

**References**

Stohl, A., Forster, C., Eckhardt, S., Spichtinger, N., Huntrieser, H., Heland, J., Schlager, H., Wilhelm, S., Arnold, F., and Cooper, O.: A backward modeling study of intercontinental pollution transport using aircraft measurements, Journal of Geophysical Research: Atmospheres, 108, https://doi.org/10.1029/2002jd002862, 2003..

Stohl, A., Forster, C., Frank, A., Seibert, P., and Wotawa, G.: Technical note: The Lagrangian particle dispersion model FLEXPART version 6.2, Atmospheric Chemistry & Physics, 5,

2461-2474, https://doi.org/10.5194/acp-5-2461-2005, 2005.

Ding, A., Wang, T., Xue, L., Gao, J., Stohl, A., Lei, H., Jin, D., Ren, Y., Wang, X., and Wei, X.: Transport of north China air pollution by midlatitude cyclones: Case study of aircraft measurements in summer 2007, Journal of Geophysical Research: Atmospheres, 114, https://doi.org/doi:10.1029/2008JD011023, 2009.

Brioude, J., Arnold, D., Stohl, A., Cassiani, M., Morton, D., Seibert, P., Angevine, W., Evan, S., Dingwell, A., Fast, J. D., Easter, R. C., Pisso, I., Burkhart, J., and Wotawa, G.: The Lagrangian particle dispersion model FLEXPART-WRF version 3.1, Geoscientific Model Development, 6, 1889-1904, https://doi.org/10.5194/gmd-6-1889-2013, 2013.

**Minor Comments**

*[1. Line 48: The order of the references is messed up, which should follow the order of the first letter of the first author for each reference alphabetically, and should be "An et al., 2019; Fuzzi et al., 2015; Nel, 2005" for this case. Please check the entire manuscript.]*

**Response 5:** We have corrected all the similar errors in the revised references.

*[2. Line 50: The definition of PM2.5 "particulate matter with an aerodynamical diameter equal to or less than 2.5 micrometers".]*

**Response 6:** It has been revised.

*[3. Line 99: change "humid environment. (see Fig. 1b)" to "humid environment (see Fig.1b)". There are so many similar typos across the entire manuscript. Please CHECK!]*

**Response 7:** We have corrected all the similar errors in the revised manuscript.

*[4. Line 101: The associated temporal variations of $PM_{2.5}$ concentrations for the study period out of ten sites in Wuhan are strongly recommended to be plotted and placed in the Supplemental.]*

**Response 8:** According to the referee's suggestions, we have added the temporal variations of $PM_{2.5}$ concentrations for the study period out of ten sites in Wuhan in the supplemental file (Fig. s2 and s3).

*[5. Line 107: Change "obviously" to "obvious".]*

**Response 9:**   It has been revised, "obviously" has been changed to "obvious".

*[6. Line 124: "heavy PM$_{2.5}$ pollution the over central-eastern China" should be revised as "heavy PM2.5 pollution over the central-eastern China".]*

**Response 10:** It has been revised.

*[7. Line 128: The number and unit should be separated (75 μg m$^{-3}$). Similar changes should be applied for the entire manuscript.]*

**Response 11:** All the similar errors have been corrected in the revised manuscript.

*[8. Line 146: "at same day." should be changed to 'at the same day,".]*

**Response 12:** "at same day." has changed to 'at the same day".

*[9. Lines 147-Line 149: The authors use "am" and "a.m." interchangeable. Please be consistent for the entire manuscript. Similar for "pm" and "p.m.".]*

**Response 13:** All the similar errors have been corrected in the revised manuscript.

*[10. Lines 161-165: Grammar error here. Please re-write this sentence. And what is the logical relationship between this sentence and the previous one? Do you try to demonstrate the reasons for this result? If so, probably it is better to begin the sentence with "There are several reasons associated with this result. Firstly, ……".]*

**Response 14:** We are so sorry for the grammar error here. Following the referee's suggestion, we have re-written the sentence as follows:

There are several reasons associated with this result. Firstly, the lower near-surface wind speed could alter the concentrations of air pollutants with a weaker advection of cold air, in conjunction with strong subsidence and stable atmospheric stratification, easily producing a stagnation area in the lower troposphere with resulting in regional pollutant accumulations for the development of haze events.

*[11. Line 165: what is "CEC" here?]*

**Response 15:** The CEC stands for Central-eastern China; and it has been revised in the manuscript.

*[12. Lines 165-170: There are many typos and grammar errors in this sentence. And I am confused by this sentence as well, which looks very odd to me. Is this your statement or conclusion? Several references to support your statement will be necessary.]*

**Response 16:** We are so sorry for the typos and grammar error here. Following the referee's suggestion, we have modified the sentence with adding the relevant references to support our statement as follows:

Secondly, in the presence of high soil moisture, strong surface evaporation could increase the near-surface relative humidity, which is also conducive to hygroscopic growth of participles for haze formation (Dawson et al., 2014; Xu et al. 2016). High air temperature and strong solar radiation could enhance chemical conversions for the formation of secondary aerosols in the atmosphere (He et al., 2012; Huang et al., 2014). Furthermore, precipitation could alter the emissions, and depositions of air pollutants (Dawson et al., 2007; Cheng et al. 2016).

**References**

Cheng, X., Zhao, T., Gong, S., Xu, X., Han, Y., Yin, Y., Tang, L., He, H., and He, J.: Implications of East Asian summer and winter monsoons for interannual aerosol variations over central-eastern China, Atmospheric Environment, 129, 218-228, https://doi.org/10.1016/j.atmosenv.2016.01.037, 2016.

Dawson, J., Adams, P., and Pandis, S.: Sensitivity of $PM_{2.5}$ to climate in the Eastern US: a modeling case study, Atmospheric chemistry and physics, 7, 4295-4309, https://doi.org/10.5194/acp-7-4295-2007, 2007.

Dawson, J. P., Bloomer, B. J., Winner, D. A., and Weaver, C. P.: Understanding the Meteorological Drivers of U.S. Particulate Matter Concentrations in a Changing Climate, Bulletin of the American Meteorological Society, 95, 521-532, https://doi.org/10.1175/bams-d-12-00181.1, 2014.

He, K., Zhao, Q., Ma, Y., Duan, F., Yang, F., Shi, Z., and Chen, G.: Spatial and seasonal variability of $PM_{2.5}$ acidity at two Chinese megacities: insights into the formation of secondary inorganic

aerosols, Atmospheric Chemistry and Physics, 12, 1377-1395, https://doi.org/10.5194/acp-12-1377-2012, 2012.

Huang, R. J., Zhang, Y., Bozzetti, C., Ho, K. F., Cao, J. J., Han, Y., Daellenbach, K. R., Slowik, J. G., Platt, S. M., Canonaco, F., Zotter, P., Wolf, R., Pieber, S. M., Bruns, E. A., Crippa, M., Ciarelli, G., Piazzalunga, A., Schwikowski, M., Abbaszade, G., Schnelle-Kreis, J., Zimmermann, R., An, Z., Szidat, S., Baltensperger, U., El Haddad, I., and Prevot, A. S.: High secondary aerosol contribution to particulate pollution during haze events in China, Nature, 514, 218-222, https://doi.org/10.1038/nature13774, 2014.

Xu, X., Zhao, T., Liu, F., Gong, S. L., Kristovich, D., Lu, C., Guo, Y., Cheng, X., Wang, Y., and Ding, G.: Climate modulation of the Tibetan Plateau on haze in China, Atmospheric Chemistry and Physics, 16, 1365-1375, https://doi.org/10.5194/acp-16-1365-2016, 2016.

*[13. Lines 184-185: There should be spaces between references, which should be "(Miao et al., 2018; Xu et al., 2016b). There are many cases (e.g., Line 187, 254, 263 and etc) like this. Please check over the entire manuscript.]*

**Response 17:** We have corrected all the similar errors in the revised manuscript.

*[14. Line 210: "the stagnation meteorological conditions" should be revised as "meteorological conditions of the stagnation".]*

**Response 18:** It has been revised as suggested by the referee.

*[15. Lines 233-234: References relevant to secondary organic and inorganic aerosols study over Wuhan?]*

**Response 19:** Following the referee's suggestion, we have modified the sentence with adding the references relevant to secondary organic and inorganic aerosols study over Wuhan as follows:

The meteorological drivers of air quality change are complicated by a series of physical and chemical processes in the atmosphere especially the formation of secondary air pollutants with strong hygroscopic growth in the humid air environment overlying the dense water network (see Fig. 1b) in the YRMB region (Cheng et al., 2014, He et al., 2012, Huang et al., 2014),

**References**

Cheng, H., Gong, W., Wang, Z., Zhang, F., Wang, X., Lv, X., Liu, J., Fu, X., and Zhang, G.: Ionic composition of submicron particles ($PM_{1.0}$) during the long-lasting haze period in January 2013 in Wuhan, central China, Journal of Environmental Sciences, 26, 810-817, https://doi.org/10.1016/s1001-0742(13)60503-3, 2014.

He, K., Zhao, Q., Ma, Y., Duan, F., Yang, F., Shi, Z., and Chen, G.: Spatial and seasonal variability of $PM_{2.5}$ acidity at two Chinese megacities: insights into the formation of secondary inorganic aerosols, Atmospheric Chemistry and Physics, 12, 1377-1395, https://doi.org/10.5194/acp-12-1377-2012, 2012.

Huang, R. J., Zhang, Y., Bozzetti, C., Ho, K. F., Cao, J. J., Han, Y., Daellenbach, K. R., Slowik, J. G., Platt, S. M., Canonaco, F., Zotter, P., Wolf, R., Pieber, S. M., Bruns, E. A., Crippa, M., Ciarelli, G., Piazzalunga, A., Schwikowski, M., Abbaszade, G., Schnelle-Kreis, J., Zimmermann, R., An, Z., Szidat, S., Baltensperger, U., El Haddad, I., and Prevot, A. S.: High secondary aerosol contribution to particulate pollution during haze events in China, Nature, 514, 218-222, https://doi.org/10.1038/nature13774, 2014.

*[16. Line 276: Change "relatively" to "relative".]*

**Response 20:** It has been changed.

*[17. Lines 296-299: First of all, there are grammar errors in this sentence (e.g., ...by winter monsoonal winds the from Tongling and Hefei to Wuhan (...). Second of all, the site numbers of Tongling and Hefei are 6 and 5 respectively, as indicated by legend of Figure 6a?)]*

**Response 21:** In the revised manuscript, the grammar errors have been corrected, and the site numbers have been modified as indicated by legend of Figure 6a.

*[18. Lines 311-313: It seems that this sentence belongs to the beginning of Section 3.2.]*

**Response 22:** Yes, this sentence (Lines 311-313) has been moved to the beginning of Section 3.2.

*[19. Lines 331-333: I recommend that the authors make a plot associated with the modeling domains, which demonstrates the regions with the coarse and finer horizontal resolutions (refer to my major comment #2).]*

**Response 23:** We have added the modeling domains with the coarse and finer horizontal resolutions in the supplemental file (Fig.s4).

*[20. Lines 341-342: I have concerns about the release of the number of particles in FLEXPART-WRF. Firstly, for particles from FLEXPART, it is not PM2.5 particles, it is just particles to represent the air parcels. Secondly, can you double check that the model release 50,000 particles per hour? From my understanding, for each hourly mean PM2.5 observation at the receptor list, the release of particles in the 48-h backward trajectory simulation in FLEXTPART just happens in the first hour, with the rest of the time tracking the routes/transport of the particles over the simulation domain?]*

**Response 24:** Yes. Many thanks for the kind review. We have carefully checked our model configuration, and accordingly corrected the errors in the revised manuscript as follows:

For particles from FLEXPART, it is not $PM_{2.5}$ particles, it is just particles to represent the air parcels, and the release of particles in the 48-h backward trajectory simulation in FLEXTPART just happens in the first hour, with the rest of the time tracking the routes/transport of the particles over the simulation domain.

*[21. Line 374: Change "Eq (1)" to "Eq. (1)".]*

**Response 25:** It has been corrected.

*[22. Lines 634-637: For "K km$^{-1}$", it should be "K km$^{-1}$".]*

**Response 26:** It has been revised.

*[23. Lines 640-645: There are many typos for Figure 1. For Y-axis title in Figure 1a, it should be "Latitude". Moreover, both units of X-axis and Y-axis in Figure 1a are missing. In Line 643, "YPD" is a typo. And where is the description of PRD here?]*

**Response 27:**Many thanks for the careful review of referee. We are sorry for the typos, which have been corrected in the revised manuscript.

*[24. Lines 663-664: The solid line for heavy $PM_{2.5}$ pollution and the dash line for clean air period are missing in the caption for Figure 5.]*

**Responses 28:** We have modified the caption for Figure 5.

*[25. Lines 679-680: Why there are no "comma" among "P1 P2 and P3". I suggest changing the caption of the last part of the caption of Figure 7 as "…. pollution periods of P1 (upper panel), P2 (middle panel) and P3 (lower panel), respectively, in January 2016".]*

**Response 29:** We are sorry for the typos. We have add a "comma" between P1 and P2 as well as modified the caption for Figure 7.

---

## Author Comment (AC2) · 21 Jan 2020

Dear Editors and Referees:

Thank you very much for your constructive suggestions and helpful comments for improving our manuscript acp-2019-758. We have accordingly made the careful revisions. Revised portions are highlighted in the revised manuscript. In the following we quoted each review question in the square brackets and added our response after each paragraph.

**Responses to Referee #2**

*[General comments.*

*In this manuscript, the authors present an observational analysis to characterize the unique features of meteorological conditions that account for the heavy air pollution events in Wu Han, a metropolis in the Yangtze River Middle Basin, China. and then use a Lagrange particle dispersion model to quantify the percentage contribution of regional transport to such heavy pollution events. They found that PM$_{2.5}$ concentrations show a positive correlation with wind speeds and no stable atmospheric boundary conditions are required to support the accumulation of air pollutants when 24-hr average PM$_{2.5}$ concentrations are higher than 150.0 $\mu g \cdot m^{-3}$. Regional transport driven by strong wind speed contributed more than 65% increase in surface PM$_{2.5}$ concentrations during the development of air pollution events in this region. The study represents a great interest to air quality community given the unique features which are very different from those presented in the textbooks. This version is improved to some extent as compared to the first submission. Part of my comments have been addressed but not all. Especially, the manuscript structure is not re-organized as suggested, a lot of grammar errors or typos need to be corrected throughout the manuscript. In addition, I have several major concerns with the authors' arguments during their analyses and discussion. Thus, a major revision is still required before it is accepted for publication. ]*

**Response 1:** Many thanks for the encouraging comments and constructive suggestions on our manuscript acp-2019-758. According to the suggestions of referee, we have re-organized

the manuscript structure with detailed descriptions of observational and modeling methods, concerns of technical methodology as well as corrected the grammar errors and typos over the entire manuscript (please find them in the following responses and the highlighted revisions in the revised manuscript).

*[Major comments*
*1. It is strongly recommended to re-organize the structure of the manuscript. Both Methodology and Results/Discussion parts are mixed together in the current version. For instance, it is suggested to move "Model Description (Section 3.2.1)", "Model Configuration (Section 3.2.2)", and the way of calculating "contribution rates" (Lines 360-375 in Section 3.3) to a new section like "Data and methods" (say Section 2 in the new version), and then move part of current Section 2.1, Sections 2.2 and 2.3 to Section 3 like "Results and Discussion" in the revised or new version something like that.]*

**Response 2:** Following the referee's suggestions, we have re-organized the structure of the paper. In the revised manuscript, the descriptions of observational data from various sites and FLEXTPART-WRF (Sect. 3.2.1,Sect. 3.2.2 and the way of calculating "contribution rates" in Section 3.3) are placed in Section 2. And Results and Discussions, including the analysis of the observational data and modeling study, are placed in Section 3 *"Results and Discussion"* .

*[2. The East Asian winter Monsoon was mentioned at least 10 times throughout the manuscript to highlight its importance in driving the regional transport during development of heavy pollution events observed in Wuhan. As we know, the East Asian Monsoon represents a seasonal mean behavior and its temporal scale is much longer than that of air pollution events which usually have a scale of one to several day(s) but not longer than one week according to the authors' argument. The authors need show some scientific evidences to support their arguments on how the East Asian winter Monsoon can drive the regional transport which may lead to the development of heavy pollution events. Otherwise the readers may get confused when they read*

*Fig.9b in where the regional transport was from East China other than North China. My suggestion is to limit the emphasis of the East Asian winter Monsoon in this study.]*

**Response 3:** We totally agree with the referee's comments and suggestions. Following the them to correct this misunderstanding on the East Asian Monsoons, we have changed "the East Asian monsoon" to "the cold air activity of East Asian winter monsoon over central-eastern China" to limit the emphasis of the East Asian winter Monsoon in this study (please see the highlighted revisions in the revised manuscript.

*[3. Estimate of percentage contribution of regional transport to the heavy pollution events in the YRMB region is one of the major works proposed by this study. As described in Eq.1 and 2, simulation of residence time of $PM_{2.5}$ is critical to do such calculations. Please define residence time. How does the FLEXPART simulate the residence time? A little bit more details are helpful for our readers to understand the percentage contribution of regional transports to the three different episodes.]*

**Response 4:** Thanks for the comments. In the revised manuscript, we have clarified the quantification of regional transport contributions with utilizing the model FLEXPART-WRF in the revised manuscript as followings:

In the model FLEXPART-WRF, the trajectory of a large number of particles released from a source is simulated with consideration of the processes of tracer transport, turbulent diffusion, wet and dry depositions in the atmosphere. With Lagrangian method, it could result in a Jacobian matrix (footprint), in unit of mass per volume per unit flux. Stohl et al. (2005) mathematically derived the residence time for particles out of FLEXPART. Generally, in the backward trajectory of FLEXPART modeling, a large number of particles is released at a receptor and transported backward in time. Then the residence time (not the lifetime) of all particles, normalized by the total number of released particles, is determined

on a uniform grid. In this study for the receptor of Wuhan, the residence time for a thickness of 100 m above the surface was calculated and considered the ''footprint'' (in unit of s). By multiplying the residence time with the air pollutant emission flux in the respective grid cell (in unit of $\mu g \ m^{-2} \ s^{-1}$ ) calculated from the Multi-resolution Emission Inventory of year 2016 for China (MEIC, http://www.meicmodel.org/), the emission source contribution (in $\mu g \ m^{-2}$) from this grid cell to the receptor could be estimated (Stohl, 2003; Stohl et al., 2005;Ding er al.,2009), yielding a so-called potential source contribution map, which is the geographical distribution of the regional transport contribution rates (%) of the emission source grid cell to $PM_{2.5}$ pollution at the receptor of Wuhan (Fig. 9).

**References**

Stohl, A., Forster, C., Eckhardt, S., Spichtinger, N., Huntrieser, H., Heland, J., Schlager, H., Wilhelm, S., Arnold, F., and Cooper, O.: A backward modeling study of intercontinental pollution transport using aircraft measurements, Journal of Geophysical Research: Atmospheres, 108, https://doi.org/10.1029/2002jd002862, 2003..

Stohl, A., Forster, C., Frank, A., Seibert, P., and Wotawa, G.: Technical note: The Lagrangian particle dispersion model FLEXPART version 6.2, Atmospheric Chemistry & Physics, 5, 2461-2474, https://doi.org/10.5194/acp-5-2461-2005, 2005.

Ding, A., Wang, T., Xue, L., Gao, J., Stohl, A., Lei, H., Jin, D., Ren, Y., Wang, X., and Wei, X.: Transport of north China air pollution by midlatitude cyclones: Case study of aircraft measurements in summer 2007, Journal of Geophysical Research: Atmospheres, 114, https://doi.org/doi:10.1029/2008JD011023, 2009.

*[4. Lines 323-329:    I assume that the FLEXPART simulations were driven by the WRF outputs rather than ECMWF or NCEP reanalysis data. If this is the case, please make clarification and delete lines 323-325.]*

**Response 5:** Following the referee's suggestion, we have made clarification with deleting lines 323-325 in the revised manuscript as follows:

In this study on the fine and multiscale modeling of air pollutant sources and regional transport, FLEXPART was coupled offline with the Weather Research and Forecasting Model (WRF) to effectively devise the combined model FLEXPART-WRF.

*[5. Fig.5b: We can see that the heavy air pollution events had stronger winds within the 1-km layer but weaker winds above the 1-km layer. Does this mean that regional transport are mainly limited to the 1-km layer. Some discussions on this will be helpful.]*

**Response 6:** Thanks for the referee's comment. We have accordingly added the following discussions in the revised manuscript:

Compared to the clean air period, the heavy air pollution events had stronger winds within the 1-km layer but weaker winds above the 1-km layer (Fig. 5b), indicating that regional transport of $PM_{2.5}$ was mainly limited to the 1-km layer, especially between 0.25 km and 0.8 km. These vertical structure of horizontal wind could conduce the downward mixing of the regionally transported air pollutants and produce the near-surface accumulations of air pollutants over the YRMB area with elevated ambient $PM_{2.5}$ concentrations, thus contributing to a heavy air pollution.

*[6. Writing needs a heavy edit work. There are a lot of grammar errors or typos and many sentences need further improvement. Some of examples include "obviously differences (L107)", "relative high (L109)", "suffering under significant (L133)", "has significantly influence (L162)", "relatively to (L276)", "a horizontally resolution (L344)", etc. I am not going to list all of them since there are many.]*

**Response 7:** Many thanks for the referee's careful review. We are so sorry for the grammar errors or typos, which have been corrected in the revised manuscript.

By the way, this revised manuscript was edited by Elsevier Language Editing Services to improve the English language.

Minor comments:
*[1. L19: central China→ Central China.]*

**Response 8:** It has been changed.

*[2. L20: I am not sure "excessive" is appropriate in this manuscript.]*

**Response 9:** The "excessive PM$_{2.5}$ concentrations" has been changed to "hourly PM$_{2.5}$ concentrations" in the revised manuscript.

*[3. L30: I did check "List of regions of China" at Wikipedia at https://en.wikipedia.org/wiki/List_of_regions_of_China, and didn't find "central-eastern China". So "Central China" should be better and sufficient.]*

Response 10: I am sorry for the geographical misleading. In the revised manuscript, we have changed *"central-eastern China" to "Central and Eastern China"* covering the major anthropogenic pollutant sources over the vast flatlands from the eastern edges of the Tibetan Plateau and the Loess Plateau to China's Pacific coast for the regional transport of air pollutants toward to YRMB.

*[4. L33: FLEXPART-WRF or WRF-FLEAPART? I would suggest the latter since it is WRF-driven FLEXPART. In addition, please define any abbreviated terms at its first appearance. Please check similar issue for other abbreviations throughout the manuscript.]*

**Response 10:** We totally agree with the referee's comments. However, the Lagrangian particle dispersion model FLEXPART-WRF was developed by Brioude et al. (2013), therefore, we adopted the model name FLEXPART-WRF in the manuscript.

All the abbreviated terms were defined at its first appearance in the revised manuscript.

**Refereces**

Brioude, J., Arnold, D., Stohl, A., Cassiani, M., Morton, D., Seibert, P., Angevine, W., Evan, S., Dingwell, A., Fast, J. D., Easter, R. C., Pisso, I., Burkhart, J., and Wotawa, G.: The Lagrangian particle dispersion model FLEXPART-WRF version 3.1, Geoscientific Model Development, 6, 1889-1904, https://doi.org/10.5194/gmd-6-1889-2013, 2013.

*[5. L155-157: Please define these abbreviations at their first appearances.]*

**Response 11:** We have defined these abbreviations at their first appearances in the revised manuscript.

[6. L251: change "the atmospheric stability in the boundary layer" to "the stability of the atmospheric boundary layer"?]

**Response 12:** It has been changed.

*[7. L272: are you sure "it is in Section 3.1"?]*

**Response 13:** It is "3. Regional transport of PM2.5 in heavy air pollution periods".

*[8. L342: Please change to (30.61ºN, 114.42ºE).]*

**Response 14:** It has been changed.

*[9. L232-235: I feel a "jump" when I read this sentence.]*

**Response 15:** We have modified this sentence in the revised manuscript as follows:

The meteorological drivers of air quality change are complicated by a series of physical and chemical processes in the atmosphere especially the formation of secondary air pollutants with strong hygroscopic growth in the humid air environment overlying the dense water network (see Fig. 1b) in the YRMB region (Cheng et al., 2014, He et al., 2012, Huang et al., 2014),

**References**

Cheng, H., Gong, W., Wang, Z., Zhang, F., Wang, X., Lv, X., Liu, J., Fu, X., and Zhang, G.: Ionic composition of submicron particles ($PM_{1.0}$) during the long-lasting haze period in January 2013 in Wuhan, central China, Journal of Environmental Sciences, 26, 810-817, https://doi.org/10.1016/s1001-0742(13)60503-3, 2014.

He, K., Zhao, Q., Ma, Y., Duan, F., Yang, F., Shi, Z., and Chen, G.: Spatial and seasonal variability of $PM_{2.5}$ acidity at two Chinese megacities: insights into the formation of secondary inorganic aerosols, Atmospheric Chemistry and Physics, 12, 1377-1395, https://doi.org/10.5194/acp-12-1377-2012, 2012.

Huang, R. J., Zhang, Y., Bozzetti, C., Ho, K. F., Cao, J. J., Han, Y., Daellenbach, K. R., Slowik, J. G., Platt, S. M., Canonaco, F., Zotter, P., Wolf, R., Pieber, S. M., Bruns, E. A., Crippa, M., Ciarelli, G., Piazzalunga, A., Schwikowski, M., Abbaszade, G., Schnelle-Kreis, J., Zimmermann, R., An, Z.,

Szidat, S., Baltensperger, U., El Haddad, I., and Prevot, A. S.: High secondary aerosol contribution to particulate pollution during haze events in China, Nature, 514, 218-222, https://doi.org/10.1038/nature13774, 2014.

*[10. L352: "The simulated meteorology" → "The simulated meteorological fields".]*

**Response 16:** It has been changed.

*[11. L373-374: Change "by calculation of the PM$_{2.5}$ contribution rates with Eq (1)" to "by using the PM$_{2.5}$ contribution rates calculated with Eq.1" something like that.]*

**Response 17:** Thanks for careful editing. We have been accordingly changed in the revised manuscript.

*[12. L309-313: I do not think this paragraph is necessary since it does not provide any useful information. Similar issue can be found in other places of the manuscript. ]*

**Response 18:** Following the referee's suggestion, we have deleted the unnecessary sentences and paragraphs.

*[13. L338-339: What are the horizontal resolutions of the NCEP reanalysis data?]*

**Response 19:** *the horizontal resolutions of the NCEP reanalysis data* is 1°×1°, which has been added in the revised manuscript.

*[14. L419-423: Does this paragraph represent any significant findings or conclusions obtained from this study? I am not sure this paragraph is really needed here.]*

**Response 20:** We have accepted the suggestion of referee and deleted this paragraph in the revised manuscript.

*[15. L424-426: We know this already and I don't think you need iterate this sentence here. It does not provide any more useful information to me.]*

**Response 21:** Following the suggestion of referee, we have deleted the sentence (L424-426) and modified the paragraph as follows:

This study of environmental and meteorological observations in the YRMB region revealed a unique "non-stagnant" meteorological condition of the boundary layer characterized by strong wind, no inversion layer and a more unstable structure in the atmospheric boundary layer associated with heavy air pollution periods with excessive $PM_{2.5}$ concentrations in the YRMB region, which facilitates understanding of the air pollutant source-receptor relationship of regional air pollutant transport. The study represents a great interest to air quality community given the unique features of air pollution meteorology which are very different from those "stagnant" meteorological conditions presented in the textbooks.

*[16.    L625-629: Please define WS, T, P, and RH in the description of Table 1 and Table 2.*

**Response 22:**    WS, T, P, and RH have been defined with wind speed, air temperature, air pressure and relative humidity respectively in the revised manuscript.

*[17. Fig.1b: The font size of those cities shown in Fig.1b is too small. Is it possible to add the locations of 10 sites presented on Page 5 at Lines 101-103 in this plot?]*

**Response 23:** We have added the locations of 10 sites in Wuhan    in the supplemental file (Fig. s1).

*[18. Fig.9: I believe that the values of the percentage contribution rates are not correct.]*

**Response 24:** Thanks for the comments. We have confirmed that the values of the percentage contribution rates in the Fig.9 are correct. Fig.9 is composed by $151 \times 161$ grid points, and the total contribution rate of all grid points is 100%.

---

## Referee Report (RR1)

Most of the comments and suggestions have been addressed in the revised version. For instance, the manuscript structure was re-organized as suggested. However, several major concerns need to be addressed before it is accepted for publication.

Major comments.

1.  Issue 1 related to manuscript-structure organization: P147-165, Section 2.2.2, the model simulation validation should be not presented here.

2.  Issue 2 related to manuscript-structure organization: the three sections, i.e., Sections 3.2, 3.3, and 3.3 are presented in the current version to illustrate the impact of meteorological conditions such as winds and stabilities of the atmospheric boundary layer on surface $PM_{2.5}$ concentrations and pollution events. I am not sure that the authors need three sub-sections to discuss them separately. To me, the observed evidence is pretty clear and straightforward that heavy $PM_{2.5}$ pollution events (daily mean concentrations higher than 150.0 $\mu g \cdot m^{-3}$) were caused by strong northly winds and light pollution was associated with local sources. I would suggest combining them or re-organizing them further in a better way.

3.  To better characterize the unique features of the ABL for the heavy pollution events in this region, the authors are suggested to shed more light on the characteristics of the ABL for different pollution events. Fig.6 is a good one but it isn't enough. For instance, which cases do the profiles presented in Fig.6 represent? What time and which day? If these profiles represented the conventional sounding data like twice a day (07am and 07 pm Beijing Time), that would be not sufficient. It will be very helpful if the authors can present any more observed profiles like LiDAR measurements, Wind Profilers data, etc. to compare the differences between the three heavy pollution events and a case with low $PM_{2.5}$ as well as their evolution. Any effort like that will be a strong support to the conclusions that the authors want to draw through this study.

4.  English writing is another major concern that the authors really need more efforts. There are many writing issues related to grammar, typo, sentence structures, inappropriate words, etc. I am not going to list them here. My suggestion is to find a professional language edit service.

---

## Author Response (AR2)

Dear Editors and Referees:

Thank you very much for your careful review and helpful comments on our manuscript acp-2019-758. We have accordingly made the careful revisions. The revised portions are highlighted in the revised manuscript. In the following we quoted each review question in the square brackets and added our response after each paragraph.

**Responses to Referee #1**

*[1. Lines 692-697: There are several typos here. "within," should be revised as period. Also, is "Thermo Fisher Scientifi" correct here? Please double check. Meanwhile, I suggest the authors add some references for using this instrument to measure PM over China.]*

**Response 1:** We have revised the typos and removed the sentence with "Thermo Fisher Scientific", following the ACP requirements to cite the instrument (instruments maker) in the manuscript. Actually, the observation data of hourly $PM_{2.5}$ concentrations in this study were under quality control that is based on China's national standard of air quality observation operated by the Ministry of Ecology and Environment (http://www.mee.gov.cn/) .

*[2. Lines 695-697: Should the "Ministry of ecology and environmental protection of China" be changed to "Ministry of Ecology and Environmental Protection of People's Republic of China"?]*

**Response 2:** It has been changed.

*[3. Line 916: change the "Meteorological" to "meteorological".]*

**Response 3:** Thanks for the careful review. It has been changed.

*[4. Phrases of "Central and Eastern China (CEC)" and "CEC" are used in the abstract. However, the authors use "Central-eastern China" in the main text. Please be consistent. I would suggest using "central-eastern China" across the entire manuscript, including the abstract.]*

**Response 4: Following the reviewer's suggestion**, we have used "CEC'' from the full Phrases of central-eastern China consistently in the abstract, the main text and the figure captions of the revised manuscript.

**Responses to Referee #2**

*[1. Issue 1 related to manuscript-structure organization: P147-165, Section 2.2.2, the model simulation validation should be not presented here.]*

**Response 1:** Many thanks for the referee's comment.

The FLEXPART model was coupled offline with the Weather Research and Forecasting (WRF) model to effectively devise the combined model FLEXPART-WRF (Fast and Easter, 2006; Brioude et al., 2013), which was used to characterize $PM_{2.5}$ sources and regional transport in this study.   The the validation of Section 2.2.2 was only for the meteorology simulated by the WRF model, Because the WRF-simulated meteorology was used to drive the FLEXPART backward trajectory simulation, it could be better to present *Section 2.2.2,* the WRF modeling meteorological validation before Section 2.3, Estimating contribution of regional transport of $PM_{2.5}$ to air pollution based on the backward trajectory of FLEXPART-WRF modeling.

By the way, we have updated the subtitle of Section 2.2.2 with "WRF modeling configuration and meteorological validation" in the revised manuscript to avoid the misleading of readers.

*[2. Issue 2 related to manuscript-structure organization: the three sections, i.e., Sections 3.2, 3.3, and 3.3 are presented in the current version to illustrate the impact of meteorological conditions such as winds and stabilities of the atmospheric boundary layer on surface PM2.5 concentrations and pollution events. I am not sure that the authors need three sub-sections to discuss them separately. To me, the observed evidence is pretty clear and straightforward that heavy PM2.5 pollution events (daily mean concentrations higher than 150.0 $\mu g \cdot m!$") were caused by strong northly winds and light pollution was associated with local sources. I would suggest combining them or re-organizing them further in a better way.]*

**Response 2:** Following the referee's suggestion, we have re-organized Section 3. Results and Discussion in the revised manuscript as follows:

1) Combining the old Section 3.1 and the first paragraph of old Section 3.2 into the new Section 3.1 with the updated subtitle of "**Variations in local PM$_{2.5}$ concentrations and meteorology in January 2016**".

**2)** Combining the the last paragraph of the old Section 3.2 and the old section 3.3 into the new Section 3.2 with the subtitle of "**A unique meteorological condition of "non-stagnation" for heavy PM$_{2.5}$ pollution**" and restructuring the new the new Section 3.2 into Section **3.2.1 Strong northerly winds and** Section **3.2.2 unstable structures in the atmospheric boundary layer.**

3) The subtitle of new Section 3.3 has been modified to "**Regional transport of PM$_{2.5}$ in northerly winds observed over CEC**".

4) The new Sections 3.4 was from the old Section 3.5.

*[3. To better characterize the unique features of the ABL for the heavy pollution events in this region, the authors are suggested to shed more light on the characteristics of the ABL for different pollution events. Fig.6 is a good one but it isn't enough. For instance, which cases do the profiles presented in Fig.6 represent? What time and which day? If these profiles represented the conventional sounding data like twice a day (07am and 07 pm Beijing Time), that would be not sufficient. It will be very helpful if the authors can present any more observed profiles like LiDAR measurements, Wind Profilers data, etc. to compare the differences between the three heavy pollution events and a case with low PM2.5 as well as their evolution. Any effort like that will be a strong support to the conclusions that the authors want to draw through this study.]*

**Response 3:** Many thanks for the referee's suggestions.

We understand that the unique features of the ABL could be better characterized for the heavy pollution events in this region with the fine data of vertical observations. However, there were no available LiDAR, Wind Profile and other observation data over the YRMB area in January 2016 for our study period, and we had to use the conventional sounding data of meteorology like twice a day (08am and 08 pm Beijing Time) to present the characteristics of the ABL structures for heavy pollution events and clean air period during January 2016. Because the heavy $PM_{2.5}$ pollution events were observed by short durations of less than 26 h from rapid accumulation to fast dissipation, we can not present the ABL vertical profiles for each heavy pollution event due to the no effective data of the conventional sounding observation, and Fig. 6 compared the vertical profiles of air temperature, wind velocity and potential temperature averaged in the heavy pollution periods P1, P2 and P3 and in the clean air period over Wuhan during January 2016 to exhibit the unique unstable ABL structures for heavy $PM_{2.5}$ pollution.

By the way, the caption of Fig. 6 has been clarified as "Fig. 6. Vertical profiles of (a) air temperature, (b) wind velocity and (c) potential temperature averaged in the heavy pollution periods P1, P2 and P3 and in the clean air period over Wuhan during January 2016." in the revised manuscript.

*[4. English writing is another major concern that the authors really need more efforts. There are many writing issues related to grammar, typo, sentence structures, inappropriate words, etc. I am not going to list them here. My suggestion is to find a professional language edit service.]*

**Response 4:** Following the referee's suggestion, the manuscript was edited by Elsevier Language Editing Services (please see the following Elsevier certificate)
.

[Figure]

**Language Editing Services**

*Registered Office:*
Elsevier Ltd
The Boulevard, Langford Lane,
Kidlington, OX5 1GB, UK.
Registration No. 331566771

**To whom it may concern**

The paper "Heavy air pollution with the unique "non-stagnant" atmospheric boundary layer in the Yangtze River Middle Basin aggravated by regional transport of PM2.5 over China" by Chao Yu was edited by Elsevier Language Editing Services.

Kind regards,

**Elsevier Webshop Support**

(This is a computer generated advice and does not require any signature)

[revised manuscript text omitted]

Wuhan.

| Period | heavy pollution period | clean air period | monthly average |
| --- | --- | --- | --- |
| | $(K\ km^{-1})$ | $(K\ km^{-1})$ | $(K\ km^{-1})$ |
| Static stability | 4.4 | 13.2 | 8.6 |
| Anomalies of stability | -4.2 | 4.6 | - |

**Table 4.** The relative contributions of regional transport over CEC to three $PM_{2.5}$ heavy pollution periods, P1, P2, and P3, in the YRMB with local contributions.

| Contribution rates | P1 | P2 | P3 | Averages |
| --- | --- | --- | --- | --- |
| Regional transport | 68.1% | 60.9% | 65.3% | 65.1% |
| Local contribution | 31.9% | 39.1% | 34.7% | 34.9% |

[Figure]

**Fig. 1.** (a) Distribution of the YRMB (orange rectangle) with the location of Wuhan (red area) and the major haze pollution regions of NCP, YRD, PRD, and SCB in CEC as well as (b) the YRMB

region with terrain height (color contours, m in a.s.l.). The river and lake network (blue areas) are downloaded from https://worldview.earthdata.nasa.gov.

[Figure]

[Figure]

**Fig. 2.** A Taylor plot with the normalized standard deviations and correlation coefficients between

WRF-simulated and observed meteorological fields. The radian of the sector represents the correlation coefficient. The solid line indicates the ratio of standard deviation between simulations and observations. The distance from the marker to "REF" reflect the normalized root-mean-square error (NRMSE).

[Figure]

**Fig. 3.** (a) Daily changes of surface $PM_{2.5}$ concentrations in Wuhan in January 2016 with $PM_{2.5}$

concentrations exceeding 75 μg m$^{-3}$ (dash line) and 150 μg m$^{-3}$ (solid lines) for light and heavy haze pollution, respectively. (b) The hourly variations of surface $PM_{2.5}$ concentrations in three heavy air pollution events, P1, P2, and P3, with excessive $PM_{2.5}$ levels (> 150 μg m$^{-3}$) marked by the shaded areas.

[Figure]

**Fig. 4.** Hourly variations of meteorological elements and PM$_{2.5}$ concentrations in Wuhan in

January 2016. Heavy air pollution periods are marked with columns in red dash lines and PM$_{2.5}$

concentrations exceeding 150 μg m$^{-3}$ (solid line in the upper panel ).

[Figure]

**Fig. 5.** A polar plot of the hourly variations in wind speed (round radius, in units of m s$^{-1}$) and direction (angles) to surface PM$_{2.5}$ concentrations (color contours, in units of μg m$^{-3}$) in Wuhan in

January 2016.

[Figure]

**Fig. 6.** Vertical profiles of (a) air temperature, (b) wind velocity and (c) potential temperature averaged in the  periods P1, P2 and P3 and  clean air period over Wuhan during January 2016.

[Figure]

**Fig. 7** (a) Distribution of the monthly averages of surface PM$_{2.5}$ concentrations observed in

January 2016 over CEC with the locations of six sites (black dots): 1. Wuhan, 2. Xinyang, 3.

Luoyang, 4. Zhengzhou, 5. Hefei, and 6. Tongling. (b) Distribution of anomalies (color contours)

of 200 m wind speeds   averaged during the three heavy air pollution periods relative to the monthly wind averages (streamlines) in January 2016 over CEC with the location of Wuhan (a light blue star).

[Figure]

[Figure]

[Figure]

**Fig. 8.** Temporal changes of PM2.5 concentrations (dotted lines) and near-surface winds (vectors)

observed at five upstream sites (Fig. 6) and Wuhan with shifts of PM2.5 peaks (marked with shaded areas) to the YRMB's heavy PM$_{2.5}$ pollution periods P1 (upper panel), P2 (middle panel) and P3

(lower panel), in January 2016.

[Figure]

**Fig. 9.** Spatial distribution of contribution rates (color contours) to PM2.5 concentrations in Wuhan with the major pathways of regional transport over CEC (dash arrows) for three heavy pollution periods (a) P1, (b) P2, and (c) P3 in January 2016 simulated by the FLEXPART-WRF model.

---

## Author Response (AR3)

Dear Editors:

Thank you very much for your careful review and helpful comments for improving our manuscript acp-2019-758. We have accordingly made the careful revisions.

Revised portions are highlighted in the revised manuscript. In the following we quoted each review question in the square brackets and added our response after each paragraph.

*[1. Page 2. Line 21. change "observational" to "observed".]*

**Response 1:** It has been changed.

*[2. Page 2. Line 35-36. remove "that contributes...YRMB area"]*

**Response 2:** It has been removed.

*[3. Page 3. Line 43. remove "traffic".]*

**Response 3:** It has been removed.

*[4. Page 3. Line 60. "As of late"? do you understand what you mean]*

**Response 4:** We have changed "As of late" to "Over recent years".

*[5. Page 4. Line 69. "The emissions factor includes ...". This statement is not correct. "chemical*

*production and transformation" should not be counted in emissions.]*

**Response 5:** We have corrected the sentence to "The emission factor includes emission source strength of air pollutants and the precursors" and moved the

"chemical transformation" into the following sentence.

*[6. Page 6. Line 106. add "of" in-between "one which".]*

**Response 6:** It has been corrected.

*[7. Page 7. Line 139. change "coupled offline with" to "driven by"]*

**Response 7:** It has been changed.

*[8. Page 8. Line 161-162. It worths to give a bit more detail on the performance for different*

*variables and sites for people who do not know much about the Taylor diagram.]*

**Response 8:** Many thanks for the editor's careful review. The performance for different variables of air temperature, wind speed, relative humidity and air pressure at five sites over CEC was validated with standard deviation and the normalized root-mean-square error between simulations and observations in the Taylor diagram.

Please the detail on the performance in the caption of Fig. 2.

*[9. Page 9. Line 169-170. Please explain how you determine the residence time.]*

**Response 9:** Many thanks for the editor's careful review. We have explained the residence time calculation as follows:

Stohl et al. (2005) mathematically derived the residence time for particles out of

FLEXPART. Generally, in the backward trajectory of FLEXPART modeling, many particles are released at a receptor and transported backward in time. Then the residence time (not the lifetime) of all particles, normalized by the total number of released particles, is determined on a uniform grid.

**Response 18:** Thanks for the article information.   It has been cited.

[revised manuscript text omitted]